# Scaling Higher-Order Graph Learning with Maximal Clique Complexes

## Abstract

Graph neural networks (GNNs) are widely used for learning on graphs but are fundamentally limited to modeling pairwise relationships. Topological models based on simplicial or cell complexes can capture higher-order structure and match or surpass the expressive power of the Weisfeiler–Leman (WL) test, but they are difficult to scale because they require constructing higher-order complexes. In this paper, we ask how to retain the expressivity of cellular Weisfeiler networks (CWNs) while improving their scalability, and how to exploit cliques efficiently on large graphs. First, we introduce simplified and factored cellular Weisfeiler–Leman (sCWL and fCWL) tests, and show that they are as expressive as the original CWL test, while achieving better scalability properties. We then define the maximal clique complex, a cell complex whose higher-order cells are the maximal cliques of the graph, and apply the corresponding simplified and factored CWNs (sCWN and fCWN) on this structure, achieving improved time and memory complexity. To avoid explicit enumeration of all maximal cliques, we propose Clique-Walk, a biased random walk that samples (maximal) cliques and scales quasi-linearly with the number of nodes. Combining maximal clique complexes with CliqueWalk yields scalable clique-based architectures that preserve CWL-level expressivity. Experiments on node and graph classification benchmarks, including large-scale datasets, show that our models are competitive with or better than GNN and higher-order baselines, while substantially reducing computational and memory costs.

## 1 Introduction

Graphs provide a natural way to represent interactions between entities, and graph neural networks (GNNs) have become the standard approach for learning on such data (Gilmer et al., 2017; Kipf & Welling, 2017; Defferrard et al., 2016). GNNs have achieved strong performance in diverse domains, including social network analysis (Fan et al., 2019), molecular property prediction (Duvenaud et al., 2015), and computer vision (Krzywda et al., 2022). However, conventional GNNs are limited to modeling pairwise interactions between nodes, which constrains their ability to capture complex multi-way relationships (Battiloro et al., 2024). To address this limitation, recent work explores higher-order structures such as simplicial complexes (Ebli et al., 2020; Bodnar et al., 2021b; Einizade et al., 2025), cell complexes (Hajij et al., 2020; Bodnar et al., 2021a), and hypergraphs (Feng et al., 2019).

Hypergraphs generalize graphs by allowing edges, called hyperedges, to connect more than two nodes (Feng et al., 2019). A hyperedge thus represents a group interaction, for example, a set of coauthors of the same paper in a co-authorship network (Wu et al., 2022). Beyond hypergraphs, cell complexes provide a general combinatorial framework that organizes higher-order structures (Hatcher, 2002). A cell complex contains cells of different dimensions: nodes (0-cells), edges (1-cells), triangles (2-cells), and so on (Bodnar et al., 2021a). Simplicial complexes are a special case of cell complexes in which all subsets of a cell are also included, ensuring closure under subset operations (Einizade et al., 2025). In this setting, entities interact whenever they differ by the addition or deletion of a single node.

Several approaches have been proposed to lift graphs into higher-order structures, allowing the use of simplicial and cell complexes for learning tasks (Bodnar et al., 2021b; Papillon et al., 2023;

Papamarkou et al., 2024). One of these strategies is the clique lifting, where simplicial or cell complexes are built by including all cliques of the graph up to a fixed size (*e.g.*, edges or triangles) (Bodnar et al., 2021a). While effective for capturing higher-order information, these methods are often computationally expensive and require significant memory resources. Furthermore, the clique problem is well-known to require algorithms with exponential runtime in the worst case (Cormen et al., 2022).

In this paper, we address two central questions: (*i*) how to simplify cellular Weisfeiler networks (CWNs) without losing expressivity, and (*ii*) how to use maximal cliques as higher-order cells in a way that scales to large graphs. To answer the first question, we introduce the simplified and factored cellular Weisfeiler–Leman (sCWL and fCWL) tests, together with their corresponding neural architectures (sCWNs and fCWNs). We show that these variants preserve the full expressive power of the original CWL test of Bodnar et al. (2021a) while achieving better scalability properties. For the second question, we propose the *maximal clique complex*, a simplified cell complex that encodes only the maximal

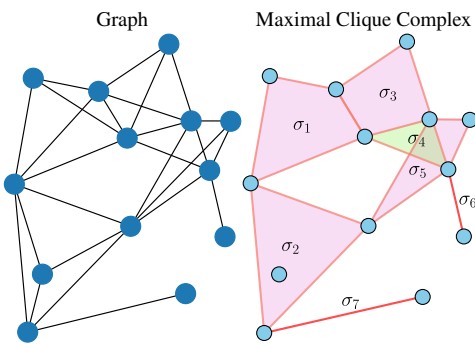

Figure 1: Maximal clique complex.

cliques of the graph (Figure 1). Because enumerating all maximal cliques can take exponential time and becomes infeasible for large graphs, we introduce CliqueWalk, a biased random-walk procedure that efficiently samples cliques and achieves quasi-linear scaling with the number of nodes. The sampled cliques define the higher-order cells used in our architectures, enabling models that achieve competitive performance on node and graph classification benchmarks while remaining scalable to large graphs.

The main contributions of this paper are:

1. We introduce the sCWL and fCWL tests and prove that they are as expressive as the regular CWL test, while offering better scaling properties.
2. We present the maximal clique complex, a simplified higher-order structure that encodes maximal cliques of a graph, and show that the resulting simplified and factored CWNs (sCWN and fCWN) are more memory- and computational-efficient than standard CWNs, without any loss in expressivity.
3. Since enumerating all maximal cliques could take exponential runtime, we propose Clique-Walk, a biased random walk algorithm that efficiently samples maximal cliques. CliqueWalk scales quasi-linearly with the number of nodes, making clique-based methods applicable to large graphs.
4. We demonstrate competitive performance on node and graph classification benchmarks. Our model matches or outperforms the accuracy of existing GNNs and topological models, while achieving substantial gains in scalability and efficiency.

## 2 RELATED WORK

The expressive power of GNNs has been extensively studied, with a particular focus on their ability to distinguish non-isomorphic graphs (Xu et al., 2019; Morris et al., 2019). It is now established that GNNs with injective aggregation functions are as powerful as the 1-WL test (Xu et al., 2019). Early architectures, such as the graph isomorphism network (GIN) (Xu et al., 2019), are explicitly designed to match the expressivity of the 1-WL test. However, GIN and related models remain limited in their ability to capture higher-order interactions (Morris et al., 2019; Bouritsas et al., 2022; Feng et al., 2022), as they rely on local message passing over pairwise connections.

To address these limitations, recent works have extended GNNs to higher-order structures. Message passing simplicial networks (Bodnar et al., 2021b) operate on simplicial complexes, exceeding the expressivity of the 1-WL test and approaching that of 3-WL. CWNs (Bodnar et al., 2021a) generalize this idea to arbitrary cell complexes, with message passing defined through boundary, co-boundary, and adjacency relations. These extensions are formalized by the CWL test, which is strictly more

expressive than 1-WL in specific cases, and have demonstrated strong empirical results, particularly in molecular graph learning (Bodnar et al., 2021a; Giusti et al., 2023).

Despite these theoretical advances, a key limitation of simplicial and cell complex models is their lack of scalability. Constructing higher-order complexes often requires enumerating large numbers of cliques, which leads to prohibitive memory and time costs. As a result, prior higher-order models, while more expressive than standard GNNs, cannot be applied efficiently to large-scale graphs. In contrast, our work introduces the maximal clique complex as a simplified higher-order structure that preserves CWL-level expressivity while enabling efficient clique-based neural architectures. Combined with our CliqueWalk sampling strategy, this provides a scalable approach to higher-order graph learning that retains strong theoretical guarantees and offers competitive performance.

## 3 PRELIMINARIES

**Notation.** Calligraphic letters denote sets, and for a set $\mathcal{X}$, $|\mathcal{X}|$ represents its cardinality. Lowercase boldface letters, like $\mathbf{x}$, represent vectors. $\bigoplus$ and COMBINE represent a mapping from a set of vectors to a vector, *e.g.*, an aggregation function.

**Cell complexes.** Cell complexes provide a natural setting for higher-order combinatorial structures.

**Definition 1** (Regular cell complex (Hansen & Ghrist, 2019; Bodnar et al., 2021a)). *A regular cell complex is a topological space $X$ that can be divided into a collection of subspaces $\{X_\sigma\}_{\sigma \in \mathcal{P}_X}$, called **cells**, where $\mathcal{P}_X$ is the set of cells induced by the topological space $X$. These cells satisfy the following properties:*

1. *Every $x \in X$ has an open neighborhood that intersects only a finite number of cells.*
2. *For any two cells $X_\sigma$ and $X_\tau$, $X_\tau \cap \overline{X_\sigma} \neq \varnothing$, if and only if $X_\tau$ is contained in $\overline{X_\sigma}$, i.e., the closure of $X_\sigma$.*
3. *Each cell is topologically equivalent (homeomorphic) to $\mathbb{R}^n$ for some dimension $n$.*
4. *For each $\sigma \in \mathcal{P}_X$, there exists a homeomorphism $\varphi$ from a closed ball in $\mathbb{R}^{n_\sigma}$ onto $\overline{X_\sigma}$, where the restriction of $\varphi$ to the interior of the ball gives a homeomorphism onto the interior of $X_\sigma$.*

A graph $G = (\mathcal{V}, \mathcal{E})$ can be interpreted as a special case of cell complexes. A graph is a one-dimensional cell where the vertices $\mathcal{V}$ and edges $\mathcal{E}$ correspond to 0-cells and 1-cells, respectively.

**Definition 2** (Cell complex adjacencies (Bodnar et al., 2021a)). *Let $X$ be a cell complex and $\sigma \in \mathcal{P}_X$ a cell. We define the following adjacency relations:*

1. *Boundary cells $\mathcal{B}(\sigma)$: lower-dimensional cells that make up the boundary of $\sigma$ (e.g., the vertices of an edge).*
2. *Co-boundary cells $\mathcal{C}(\sigma)$: higher-dimensional cells for which $\sigma$ is part of their boundary (e.g., an edge incident to a vertex).*
3. *Lower adjacent cells $\mathcal{N}_\downarrow(\sigma)$: cells of the same dimension as $\sigma$ that share at least one boundary cell with it (e.g., edges that meet at a common vertex).*
4. *Upper adjacent cells $\mathcal{N}_\uparrow(\sigma)$: cells of the same dimension as $\sigma$ that both lie on the boundary of a higher-dimensional cell (e.g., two vertices that are connected by an edge).*

**WL test.** A key challenge in graph theory is the graph isomorphism problem, which concerns deciding whether two graphs have the same structure. Finding exact solutions is often computationally demanding, so faster approximate techniques, such as graph hashing, are commonly employed. A classical and widely used technique for graph isomorphism test is the WL test (Leman & Weisfeiler, 1968). The WL test provides an efficient heuristic for the graph isomorphism problem. The formal definition of the WL test is provided in Appendix A. Beyond graphs, it can be naturally extended to regular cell complexes, capturing richer combinatorial structures.

**CWL test.** The adjacency relations in Definition 2 allow us to define the CWL scheme, which generalizes the WL test from graphs to higher-dimensional cell complexes.

**Definition 3** (CWL scheme (Bodnar et al., 2021a)). *Let $X$ be a regular cell complex. The CWL scheme is defined as follows:*

1. *Initialization: All cells $\sigma \in \mathcal{P}_X$ are assigned the same initial color.*

2. *Color refinement: At iteration $t + 1$, the color of each cell $\sigma$ is updated according to $c_\sigma^{t+1} = HASH(c_\sigma^t, c_{\mathcal{B}(\sigma)}^t, c_{\mathcal{C}(\sigma)}^t, c_{\mathcal{N}_\downarrow(\sigma)}^t, c_{\mathcal{N}_\uparrow(\sigma)}^t)$, where HASH is an injective function that combines the current color of $\sigma$ with the colors of its boundary, co-boundary, and adjacent cells.*

3. *Termination: The process is repeated until the coloring stabilizes. Two cell complexes are considered non-isomorphic if their color histograms differ.*

The CWL test is invariant under cell-complex isomorphisms. Given a map from a graph to a cell complex that preserves isomorphisms, we can use the CWL test to check graph isomorphism. This is exactly what Bodnar et al. (2021a) called a cellular lifting map (their Definition 8). Similarly, we can relate CWL to WL test in the case of skeleton preserving lifting map:

**Definition 4** (Skeleton preserving lifting (Bodnar et al., 2021a)). *A lifting map $f(\cdot)$ is skeleton-preserving if for any graph $G = (\mathcal{V}, \mathcal{E})$: (i) $f(G)$ contains $\mathcal{V}$ and $\mathcal{E}$ as cells, and (ii) the cell complex $f(G)$ restricted to node and edge set is isomorphic to $G$, i.e., the incidence matrix of $G$ and $f(G)$ are equal with the correct permutation.*

The CWL scheme has been proven to be more expressive than the standard WL test (Bodnar et al., 2021a) for skeleton preserving lifting maps. In the following, we introduce a new cell test that does not require the skeleton preserving lifting map to be more expressive than the WL test. We also, introduce the specific structures that will be the focus of our study. All proofs of theorems and propositions are provided in Appendix B.

## 4 SCALING CELL COMPLEX MODELS AND MAXIMAL CLIQUES

### 4.1 CELL COMPLEX EXPRESSIVITY THEORY

Bodnar et al. (2021a) shows we can simplify the CWL test while retaining the same expressivity.

**Theorem 5** (Bodnar et al. (2021a)). *The CWL update rule restricted to boundary and upper adjacency messages is equivalent in expressive power to the full CWL update rule.*

We also demonstrate that a different simplified version retains the same expressivity.

**Theorem 6.** *The CWL update rule restricted to boundary and co-boundary messages, called the simplified CWL (sCWL) test, is equivalent in expressive power to the full CWL update rule.*

This restricted scheme is useful in practice, as it leads to more computationally efficient models.

We also introduce a new test on cell complexes that, while keeping the node structure, enables at least the same expressivity as the CWL, sCWL, and WL tests.

**Definition 7** (Factored CWL (fCWL) test). *Let $(\mathcal{G}, \mathcal{X})$ be a graph and a cell complex constructed from a cellular lifting map that preserves the node set. The fCWL scheme is defined as follows:*

1. *Initialization: All cells are assigned the same initial color.*

2. *Color refinement: At iteration $t + 1$, the color of each non node cells $\sigma$ is updated according to $c_\sigma^{t+1} = HASH(c_\sigma^t, c_{\mathcal{B}(\sigma)}^t, c_{\mathcal{C}(\sigma)}^t)$. The color of each node $i$ is updated according to $c_i^{t+1} = HASH(c_i^t, c_{\mathcal{C}(i)}^t, c_{\mathcal{N}(i)}^t)$,*

3. *Termination: The process is repeated until the coloring stabilizes. Two cell complexes are considered non-isomorphic if their color histograms differ.*

**Theorem 8.** *fCWL is at least as expressive as WL and CWL.*

We use the ideas from sCWL and fCWL tests to introduce cellular neural networks with the same guarantees and better scaling properties than CWN (Bodnar et al., 2021a).

### 4.2 NEURAL NETWORK MODELS

We now describe several neural network architectures based on the CWL framework. These models perform message passing along the hierarchical structure of cells, propagating information through boundary, co-boundary, and adjacency relations.

**Definition 9** (CWNs). *Following (Bodnar et al., 2021a, Section 4), CWNs aggregate messages along both upper adjacency and boundary relations (Theorem 5). For a cell $\sigma$, the updates are defined as:*

$$\mathbf{m}_{\uparrow}(\sigma) = \bigoplus_{\tau \in \mathcal{N}_{\uparrow}(\sigma), \delta \in \mathcal{C}(\sigma) \bigcap \mathcal{C}(\tau)} M_{\uparrow}(\mathbf{x}_{\sigma}, \mathbf{x}_{\tau}, \mathbf{x}_{\delta}), \quad \mathbf{m}_{\mathcal{B}}(\sigma) = \bigoplus_{\tau \in \mathcal{B}(\sigma)} M_{\mathcal{B}}(\mathbf{x}_{\sigma}, \mathbf{x}_{\tau}), \tag{1}$$

$$\mathbf{x}_{\sigma} \leftarrow COMBINE(\mathbf{x}_{\sigma}, \mathbf{m}_{\mathcal{B}}(\sigma), \mathbf{m}_{\uparrow}(\sigma)), \tag{2}$$

*where $\mathbf{x}_{\sigma}$ the features of cell $\sigma$. We write $\mathbf{m}_{\uparrow}(i)$ for the aggregated message to cell $\sigma$ from all tuples formed by $\sigma$, one of its upper neighbors, and a parent they share. Similarly, $\mathbf{m}_{\mathcal{B}}(\sigma)$ denotes the aggregated message to cell $\sigma$ from all of its children.*

We now introduce a model that scales more efficiently.

**Definition 10** (Simplified CWNs (sCWN)). *Based on the restricted CWL update using only boundary and co-boundary messages (Theorem 6), we define a simplified message passing scheme:*

$$\mathbf{m}_{\mathcal{C}}(\sigma) = \bigoplus_{\tau \in \mathcal{C}(\sigma)} M_{\mathcal{C}}(\mathbf{x}_{\sigma}), \quad \mathbf{m}_{\mathcal{B}}(\sigma) = \bigoplus_{\tau \in \mathcal{B}(\sigma)} M_{\mathcal{B}}(\mathbf{x}_{\tau}), \tag{3}$$

$$\mathbf{x}_{\sigma} = COMBINE(\mathbf{x}_{\sigma}, \mathbf{m}_{\mathcal{C}}(\sigma), \mathbf{m}_{\mathcal{B}}(\sigma)), \tag{4}$$

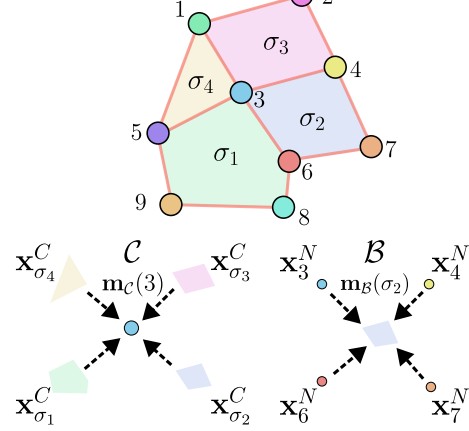

Figure 2 shows an example of the aggregation functions in Definition 10. This simplified variant reduces computational and memory requirements while retaining the expressive power of the full CWL update. Messages are propagated only along boundary and co-boundary relations, making sCWN efficient for large complexes (see Proposition 14).

Figure 2: Illustration of sCWN.

We also introduce a cell model which has a complexity between sCWN and CWN, but has better expressivity guarantees (Theorem 8, Proposition 14). We use both the clique structure and the neighborhood structure from the graph.

**Definition 11** (Factored CWNs (fCWN)). *fCWN aggregate messages using both cell complex structure and graph structure:*

$$\mathbf{m}_{\mathcal{C}}(\sigma) = \bigoplus_{\tau \in \mathcal{C}(\sigma)} M_{\mathcal{C}}(\mathbf{x}_{\sigma}, \mathbf{x}_{\tau}), \quad \mathbf{m}_{\mathcal{B}}(\sigma) = \bigoplus_{\tau \in \mathcal{B}(\sigma)} M_{\mathcal{B}}(\mathbf{x}_{\sigma}, \mathbf{x}_{\tau}), \quad m_{\mathcal{N}}(i) = \bigoplus_{j \in \mathcal{N}(i)} M_{\mathcal{N}}(\mathbf{x}_i, \mathbf{x}_j) \tag{5}$$

$$\mathbf{x}_i \leftarrow COMBINE(\mathbf{x}_i, \mathbf{m}_{\mathcal{C}}(i), \mathbf{m}_{\mathcal{N}}(i)), \quad \mathbf{x}_{\sigma} \leftarrow COMBINE(\mathbf{x}_{\sigma}, \mathbf{m}_{\mathcal{B}}(\sigma), \mathbf{m}_{\mathcal{C}}(\sigma)). \tag{6}$$

This model has better memory and time complexity than CWN in practical cases and has better expressivity guarantees (Theorem 8).

Under certain constraints, we can provide expressivity guarantees for these models.

**Proposition 12.** *sCWN and CWN are at most as expressive as CWL. If they use injective aggregation, they are equally expressive as CWL. fCWN with injective aggregation is at least as expressive as CWL and WL.*

We can further scale the sCWN and fCWN models to large graphs while still exploiting higher-order topology. We propose to use clique-based cell complexes, where maximal cliques serve as higher-dimensional cells that compactly summarize multiple nodes and edges.

### 4.3 CLIQUEWALK

**Definition 13** (Maximal clique complex). *Given a graph $G = (\mathcal{V}, \mathcal{E})$, the **maximal clique complex** is defined as:*

   *1. The 0-cells correspond to the vertices $\mathcal{V}$ of $G$.*

2. *The higher-dimensional cells correspond to the maximal cliques of $G$.*

*The set of non $0$-cells (maximal cliques) is denoted as $\mathcal{X}$.*

An example of a maximal clique complex constructed from a graph is shown in Figure 1. If we impose closure under subset operations, the maximal clique complex becomes the *clique complex* (Kahle, 2009), that is, the simplicial complex induced by including all subsets of each clique.

**Proposition 14.** *The time and memory complexities of the different CWN variants on maximal clique complexes are as follows:*

- *CWN has time and memory complexity $\mathcal{O}(n + \sum_{\sigma \in \mathcal{X}} |\sigma|^2)$.*
- *fCWN has time complexity $\mathcal{O}(|\mathcal{E}| + \sum_{\sigma \in \mathcal{X}} |\sigma|)$ and memory complexity $\mathcal{O}(n + \sum_{\sigma \in \mathcal{X}} |\sigma|)$.*
- *sCWN has time complexity $\mathcal{O}(\sum_{\sigma \in \mathcal{X}} |\sigma|)$ and memory complexity $\mathcal{O}(n + |\mathcal{X}|)$.*

*Here, $n$ is the number of nodes, $\mathcal{X}$ is the set of maximal cliques, and $\mathcal{E}$ is the set of edges. A table of all complexities can be found in the Appendix B.3.*

**Remark 15.** *These models can be simplified to reduce time and memory, for example, by using only incoming information during aggregation. These simplified versions keep the same theoretical expressivity but may capture less complex interactions between cells.*

**Remark 16.** *We conjecture that CWL on maximal cliques is more expressive than WL. A discussion of its expressive power is provided in Appendix C.*

Identifying all maximal cliques in a graph is computationally infeasible, as the clique enumeration problem might have exponential runtime.

**Proposition 17** (Moon & Moser (1965)). *A graph with $n$ nodes can contain up to $3^{n/3}$ maximal cliques.*

To circumvent this challenge, we propose a biased random walk method for efficient clique sampling, which we refer to as CliqueWalk. Our approach is inspired by existing clique sampling strategies (Bron & Kerbosch, 1973; Tomita et al., 2006; Cazals & Karande, 2008). The key idea is to grow cliques incrementally while maintaining an efficient lookup of candidate nodes that can extend the current clique, continuing until no further extension is possible. The method is summarized in Algorithm 1, and is illustrated in Figure 3. CliqueWalk enables us to sample a representative subset of cliques without exhaustively enumerating all of them. A comparison with other clique sampling schemes can be found in Appendix D.

---

**Algorithm 1** CliqueWalk

1: **procedure** CLIQUEWALK(node $i$, neighbor map $\mathcal{N}$, max walk size $\omega_{max}$)
2:     Walk $\leftarrow [i]$
3:     neighbor $\leftarrow \mathcal{N}_i$
4:     **while** neighbor $\neq \varnothing$ and $|\text{Walk}| < \omega_{max}$ **do**
5:         Choose $j \in$ neighbor
6:         Append $j$ to Walk
7:         neighbor $\leftarrow$ neighbor $\cap \mathcal{N}_j$
8:     **end while**
9:     **return** Walk
10: **end procedure**

---

**Proposition 18.** *If $\omega_{max} > \omega(G)$, each random walk generated by CliqueWalk produces a maximal clique of the graph. Where $\omega(G)$ is the maximum clique size and $\omega_{max}$ the maximum walk length.*

We denote our random walk method as CliqueWalk($n_{\text{walk}}, \omega_{\text{max}}$), where $n_{\text{walk}}$ specifies the number of walks sampled per node and $\omega_{\text{max}}$ corresponds to maximum size of walks.

**Proposition 19.** *The time complexity of CliqueWalk($n_{walk}, \omega_{max}$) on a graph $G$ is $\mathcal{O}(n \cdot n_{walk} \cdot d_{max}(G) \cdot \max(\omega(G), \omega_{max}))$, where $n$ is the number of nodes, $d_{max}(G)$ is the maximum node degree, and $\omega(G)$ is the size of the largest clique in $G$.*

The motivation for using CliqueWalk in learning is that enumerating all cliques is computationally prohibitive for large graphs. By sampling a sufficiently large number of cliques, we can approximate the local clique structure effectively. This approach allows models to capture higher-order structural information efficiently, while achieving performance comparable to, or even better than, full clique enumeration. Empirical results supporting these claims are presented in the next section.

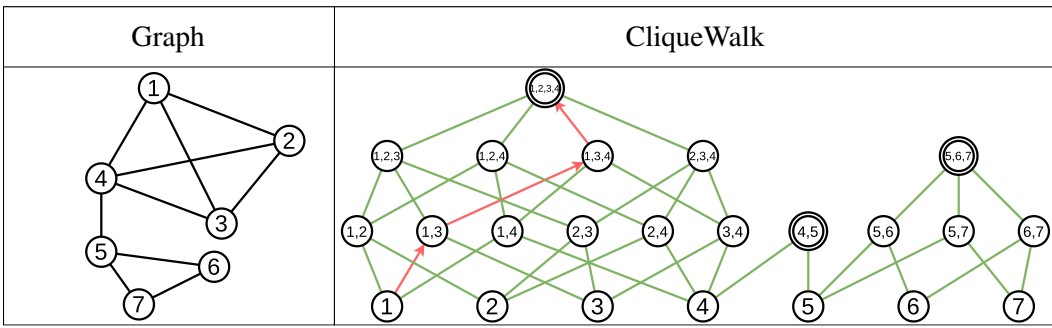

Figure 3: Illustration of a CliqueWalk starting at node 1. CliqueWalk starts from node 1 and grows a clique by repeatedly sampling a node that is adjacent to all nodes in the current clique; in the example, it successively adds nodes 3, 4, and 2 to reach the maximal clique $\{1, 2, 3, 4\}$.

## 5 EXPERIMENTS AND RESULTS

In this section, we describe the datasets and experimental setups used for node and graph classification tasks. We compare our sCWN model with: GCN (Kipf & Welling, 2017), GIN (Xu et al., 2019), SAGEConv (Hamilton et al., 2017), SGC (Wu et al., 2019), HGNN (Feng et al., 2019), SCCN (Yang et al., 2022), CWN (Bodnar et al., 2021a). We additionally present a sensitivity analysis to assess the robustness of our methodology, along with an ablation study.

### 5.1 DATASETS

**Node classification datasets.** We evaluate our models on two topological datasets (contact-primary-school and contact-high-school) (Chodrow et al., 2021; Mastrandrea et al., 2015), three citation networks (Citeseer, Cora, and PubMed) (Sen et al., 2008; Namata et al., 2012), and purchase networks like Amazon Photo network (McAuley et al., 2015; Shchur et al., 2018) and OGBN-Products (OGBN-P) (Bhatia et al., 2016). In addition, we propose a new synthetic dataset, the *stochastic clique model* (SCM), derived from the stochastic block model (SBM).

**Stochastic clique model.** It is a special case of SBM (Holland et al., 1983) where inward probability is set to 1. Graphs are generated by assembling cliques, with nodes inside each clique fully connected. Each clique is assigned a label, which is inherited by all its nodes, and node features are generated from a Gaussian distribution with a mean determined by the node label and a fixed diagonal variance. To introduce topological noise, each node is connected to nodes outside its clique with a fixed probability, perturbing the clique structure. The task can thus be interpreted as a form of label denoising.

**Graph classification datasets.** We perform experiments on two social network datasets (IMDB-BINARY, IMDB-MULTI) (Yanardag & Vishwanathan, 2015) and four molecular datasets (MUTAG, PROTEINS, NCI1, NCI109) (Borgwardt et al., 2005; Schomburg et al., 2004; Dobson & Doig, 2003a; Wale et al., 2008; Shervashidze et al., 2010) from the TUDataset (Morris et al., 2020).

**Synthetic cliques.** To compare the inference time and memory footprint of clique-based methods, we also construct a synthetic dataset of isolated cliques. This dataset allows us to systematically evaluate the computational scaling of CWN models with respect to clique size.

### 5.2 EXPERIMENTAL SETUP

**Experiments.** For node classification, we hold out 20% of the nodes as a final test set, which is used only once for reporting the final performance. The remaining 80% of the nodes are further split into 60% for training, 20% for validation, and 20% for an internal test set used during hyperparameter optimization. During training, we select the model checkpoint that achieves the highest validation accuracy and report its accuracy on the final test set. In OGBN-Products, we use the public splits and do not perform hyperparameter optimization. In graph classification, we follow the experimental protocol of Xu et al. (2019). Specifically, we perform 10-fold cross-validation on all datasets, report

Table 1: Node classification accuracy (%) with standard deviation. Best results are in **bold**, second best are underlined. HighSchool = contact-high-school, PrimarySchool = contact-primary-school. ♦ GNNs, ♣ simplicial neural networks, ♠ hypergraph neural networks, ⌘ CWN, and ★ CWNs (ours). Statistical significance: * $p < 0.05$, *** $p < 0.001$ (Welch's t-test against the best model).

| Model | Citeseer | Cora | Photo | PubMed | HighSchool | PrimarySchool | SCM | OGBN-P |
|---|---|---|---|---|---|---|---|---|
| ♦ GCN | $\mathbf{73.7_{\pm 0.76}}$ | $\mathbf{88.7_{\pm 0.61}}$ | $93.9^{***}_{\pm 0.27}$ | $88.3^{***}_{\pm 0.33}$ | $98.2^{*}_{\pm 2.6}$ | $88.9^{*}_{\pm 3.1}$ | OOM | $70.4^{***}_{\pm 0.2}$ |
| ♦ GAT | $72.2^{***}_{\pm 1.3}$ | $87.5^{***}_{\pm 1.2}$ | $93.7^{***}_{\pm 0.26}$ | $87.2^{***}_{\pm 0.33}$ | $\underline{19.1^{***}_{\pm 7.3}}$ | $13.9^{***}_{\pm 7.8}$ | OOM | OOM |
| ♦ GIN | $69.3^{***}_{\pm 1.1}$ | $86.2^{***}_{\pm 0.62}$ | $88.0^{***}_{\pm 2.2}$ | $86.7^{***}_{\pm 0.42}$ | $94.5^{***}_{\pm 3.5}$ | $85.9^{***}_{\pm 4.6}$ | OOM | $76.4^{***}_{\pm 0.4}$ |
| ♦ SAGEConv | $72.4^{***}_{\pm 1.2}$ | $\mathbf{88.7_{\pm 0.99}}$ | $95.0^{*}_{\pm 0.29}$ | $89.5_{\pm 0.6}$ | $14.6^{***}_{\pm 4.2}$ | $6.53^{***}_{\pm 4.5}$ | OOM | $78.5^{***}_{\pm 0.3}$ |
| ♦ SGC | $\mathbf{73.7_{\pm 0.74}}$ | $88.4_{\pm 0.86}$ | $89.8^{***}_{\pm 0.39}$ | $89.2^{***}_{\pm 0.21}$ | $6.3^{***}_{\pm 4.1}$ | $3.57^{***}_{\pm 3.0}$ | $65.6^{***}_{\pm 0.01}$ | $76.1^{***}_{\pm 0.07}$ |
| ♣ SCCN | $46.4^{***}_{\pm 1.4}$ | $64.4^{***}_{\pm 1.9}$ | $64.8^{***}_{\pm 2.6}$ | $73.4^{***}_{\pm 0.7}$ | $93.0^{***}_{\pm 2.5}$ | $74.1^{***}_{\pm 3.7}$ | OOM | OOM |
| ♠ HGNN | $72.9^{***}_{\pm 1.1}$ | $88.5_{\pm 0.9}$ | $94.2^{***}_{\pm 0.5}$ | $88.5^{***}_{\pm 0.39}$ | $95.4^{***}_{\pm 3.8}$ | $80.4^{***}_{\pm 5.3}$ | $68.1^{***}_{\pm 0.3}$ | $63.5^{***}_{\pm 1.0}$ |
| ⌘ CWN | $72.0^{***}_{\pm 1.6}$ | $81.1^{***}_{\pm 1.0}$ | $94.7^{***}_{\pm 0.37}$ | $89.3^{***}_{\pm 0.35}$ | $94.6^{***}_{\pm 2.2}$ | $\mathbf{90.7_{\pm 1.9}}$ | OOM | OOM |
| ★ fCWN | $72.5^{*}_{\pm 1.4}$ | $88.1^{*}_{\pm 0.79}$ | $95.1_{\pm 0.35}$ | $89.4^{*}_{\pm 0.31}$ | $\mathbf{99.5_{\pm 0.9}}$ | $\underline{89.5_{\pm 2.3}}$ | OOM | $\mathbf{78.8_{\pm 0.2}}$ |
| ★ sCWN | $\underline{72.9^{*}_{\pm 1.3}}$ | $87.3^{***}_{\pm 0.87}$ | $\mathbf{95.3_{\pm 0.39}}$ | $\mathbf{89.7_{\pm 0.35}}$ | $96.0^{***}_{\pm 2.4}$ | $86.4^{***}_{\pm 4.4}$ | $\mathbf{77.7_{\pm 0.05}}$ | $71.6_{\pm 0.5}$ |

the mean accuracy across folds at each epoch, and select the epoch with the highest mean accuracy for final evaluation.

**Implementation details.** In all experiments, we use the same architecture and swap only the convolution module for the method under evaluation. Each model is trained both with and without batch normalization, and we report results using the configuration that performs best. For all cell and hypergraph models on node or graph classification, we use the CliqueWalk lifting procedure with 8 walks per node, and initialize clique features using clique length. Cliques are sampled once and then kept fixed throughout training (no resampling). We select 8 walks as this provides a good tradeoff between accuracy and runtime across datasets. No further hyperparameter tuning regarding CliqueWalk is performed to ensure fair comparison.

For node classification, except OGBN-Products, we perform a grid search over learning rate $\{10^{-2}, 10^{-3}\}$, number of layers $\{2, 4\}$, hidden dimension $\{32, 64\}$, dropout $\{0, 0.2, 0.5\}$ and with or without BatchNorm for all models. For *contact-school* datasets, we also include GraphNorm[1] (Cai et al., 2020). Models are trained for 200 epochs on standard datasets and 500 epochs on topological ones,[2] with each grid search repeated five times using different random seeds. Final evaluation is based on 20 independent runs with new seeds. For OGBN-Products, we use fixed hyperparameters (see Appendix F) and train for 1000 epochs. For graph classification, all models use five layers (including the input convolution) and a hidden dimension of 64, while grid search is limited to dropout $\{0, 0.5\}$, batch size $\{32, 128\}$, and with or without BatchNorm.

### 5.3 RESULTS AND DISCUSSION

**Node classification.** Table 1 reports the results for the node classification task. The SCM dataset contains approximately 6M nodes and 276M edges, making it significantly larger and more challenging than standard benchmarks. In this specific case, we only use 1 random walks in CliqueWalk. Additional statistics for all datasets are provided in Table 7 in Appendix F. On topological datasets like *contact-high-school* and *contact-primary-school*, topological models have competitive performance, while classical GNNs with GraphNorm can match or exceed their performance, with fCWN still slightly better. On citation benchmarks like Citeseer and Cora, differences are small, showing no clear advantage for topological methods. On OGBN-Products, fCWN slightly outperforms SAGEConv. Unlike other higher-order methods that run out of memory (OOM), our fCWN and sCWN models scale efficiently.

**Graph classification.** Table 2 summarizes the results of the graph classification task. On social network datasets such as IMDB-B and IMDB-M, topological models achieve good performance, consistent with prior work (Bodnar et al., 2021a). In contrast, on molecular datasets, their performance is generally lower, suggesting that clique-based features are less informative for chemical graph structures. This discrepancy highlights that the benefits of higher-order information are

---

[1]The estimation of the statistics with BatchNorm on small datasets degrades model performance.

[2]GNNs converge more slowly on topological datasets, hence the larger number of epochs.

Table 2: Graph classification accuracy (%) with standard deviation. Best results are in **bold**, second best are underlined. ♦ GNNs, ♠ hypergraph neural networks, ⌖ CWN, and ★ CWNs (ours).

| Model | IMDB-B | IMDB-M | MUTAG | NCI1 | NCI109 | PROTEINS |
|---|---|---|---|---|---|---|
| ♦ GCN | $74.3_{+4.6}$ | $52.4_{+4.1}$ | $84.1_{+8.8}$ | $80.4_{+1.8}$ | $76.9^*_{+1.7}$ | $77.0_{+5.1}$ |
| ♦ GAT | $74.8_{+3.0}$ | $51.6_{+3.7}$ | $84.6_{+8.6}$ | $79.6^*_{+3.1}$ | $\underline{73.8^{***}_{+1.3}}$ | $76.5_{+3.2}$ |
| ♦ GIN | $72.1^*_{+3.8}$ | $49.7^*_{+3.4}$ | $\mathbf{89.4_{+7.8}}$ | $\underline{80.8_{+2.1}}$ | $74.8^{***}_{+2.4}$ | $75.8_{+3.4}$ |
| ♦ SAGEConv | $74.3_{+4.1}$ | $\mathbf{52.9_{+4.0}}$ | $84.6_{+9.5}$ | $\mathbf{81.5_{+1.8}}$ | $\mathbf{78.0_{+1.5}}$ | $76.3_{+4.5}$ |
| ♠ HGNN | $\mathbf{75.5_{+4.3}}$ | $52.3_{+4.8}$ | $86.2_{+8.2}$ | $79.2^*_{+3.1}$ | $76.2^*_{+1.9}$ | $76.5_{+3.9}$ |
| ⌖ CWN | $66.0^{***}_{+7.8}$ | $50.5^*_{+3.4}$ | $85.1_{+7.3}$ | $63.7^{***}_{+1.9}$ | $63.1^{***}_{+2.0}$ | $\underline{77.0_{+3.4}}$ |
| ★ fCWN | $71.9^*_{+4.1}$ | $52.8_{+2.6}$ | $85.1_{+8.1}$ | $79.2^*_{+2.4}$ | $62.3^{***}_{+4.5}$ | $75.9_{+3.3}$ |
| ★ sCWN | $\underline{75.0_{+4.5}}$ | $\underline{52.3_{+4.2}}$ | $85.7_{+8.2}$ | $66.3^{***}_{+8.9}$ | $64.1^{***}_{+2.8}$ | $\mathbf{77.5_{+3.5}}$ |

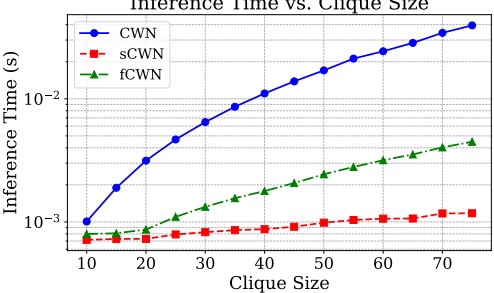

(a) Inference time with growing cliques.

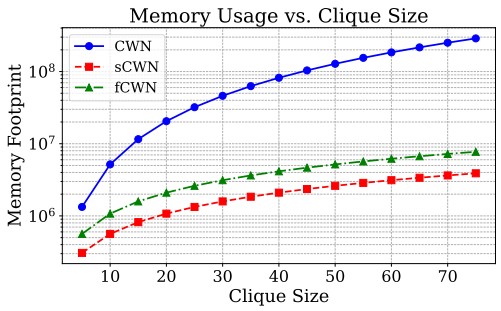

(b) Memory footprint with growing cliques.

Figure 4: Comparison of CWN, sCWN, and fCWN models with increasing clique size: (a) inference time, (b) memory footprint in number of elements in memory.

domain-dependent: social networks naturally contain larger and more meaningful cliques, whereas molecular graphs are often dominated by small motifs such as functional groups, where clique information seems to provide less meaningful information.

**Remark 20.** *Across both node and graph classification, topological models perform better on datasets with larger cliques. Table 7 in Appendix F reports the average clique size of each dataset, showing a clear correlation between larger cliques and stronger performance of topological models.*

### 5.4 SENSITIVITY ANALYSIS AND ABLATION STUDY

**Scalability of CWN models.** Figure 4 illustrates how CWN, fCWN, and sCWN scale with increasing clique size. Consistent with Proposition 14, both fCWN and sCWN require substantially less memory and runtime than CWN. Among them, sCWN achieves the best efficiency, confirming that restricting message passing to boundary and co-boundary relations provides a favorable tradeoff between expressivity and computational cost.

**Sampling effect for CliqueWalk.** We compare exact enumeration of maximal cliques with Clique-Walk sampling using between 1 and 256 walks per node (Figure 5a). A clear pattern emerges: sCWN and fCWN maintain consistent accuracy across different clique sampling rates. This demonstrates that subsampling maximal cliques via CliqueWalk reduces inference time while preserving performance. We observe a slightly different effect on smaller datasets like *contact-primary-school* as shown in Appendix E.

**CliqueWalk compute time.** We compare CliqueWalk with 8 and 64 walks against exact clique enumeration and triangle-based simplicial complex lifting (Figure 5b). Across all datasets, CliqueWalk consistently achieves substantially lower runtimes. Even with 64 walks per node, it remains close to an order of magnitude faster than both exact clique computation and simplicial lifting, while maintaining competitive accuracy. These results highlight the efficiency and scalability of the method, showing that CliqueWalk can provide a practical alternative to more costly exact approaches.

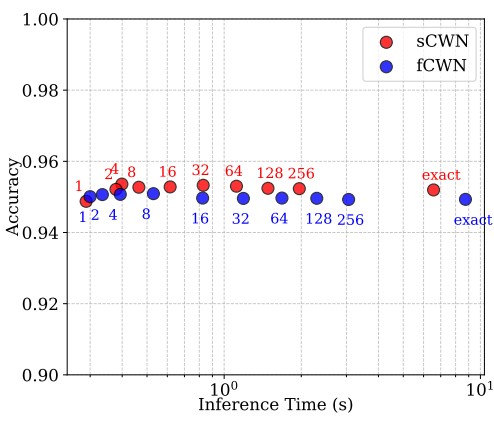 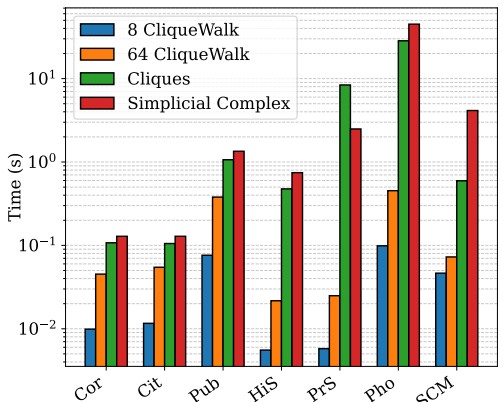

(a) Accuracy of sCWN and fCWN at different CliqueWalk sampling rates on Photo.

(b) Computation time of different lifting strategies measured on an NVIDIA RTX 3090 GPU.

Figure 5: Sensitivity analysis of CliqueWalk. (a) Accuracy as a function of the number of sampled walks. (b) Runtime comparison between CliqueWalk and exact lifting methods. Cor = Cora, Cit = Citeseer, Pub = PubMed, HiS = contact-high-school, PrS = contact-primary-school, Pho = Photo.

**Ablation study, resampling in CliqueWalk.** Table 3 compares the performance when using 8 walk CliqueWalk with or without re-sampling at each training epoch on the *contact-primary-school* and *Photo* datasets. We observe that the results are slightly better across both datasets when re-sampling, while it introduces a slight increase in run-

Table 3: Ablation sampling CliqueWalk.

| Strategy | PrimarySchool | Photo |
|---|---|---|
| Re-sampling | $87.1_{\pm3.8}$ | $95.4_{\pm0.44}$ |
| No re-sampling | $86.4_{\pm4.4}$ | $95.3_{\pm0.39}$ |

time (see PrS and Pho in Figure 5b). This suggests that using re-sampling can be a nice way to trade better generalization against computational cost.

## 5.5 LIMITATIONS

While our work establishes a scalable framework for clique-based higher-order learning, it has some limitations. First, we restrict our evaluation to node and graph classification tasks; extending the approach to other settings, such as hyperedge prediction, link prediction, or generative modeling, remains an open direction. Second, our method does not explicitly expand the receptive field of nodes, and thus may not fully capture long-range dependencies compared to approaches that incorporate multi-hop information. Finally, we focus exclusively on clique-based sampling strategies, whereas exploring alternative lifting procedures or hybrid strategies could further improve efficiency and generalization. Addressing these limitations offers promising avenues for future research.

## 6 CONCLUSION

We introduced the maximal clique complex as a simplified higher-order structure that connects clique-based representations to the CWL test, and showed that a sCWN operating on this complex achieves CWL-level expressivity while remaining computationally efficient. To address scalability, we proposed CliqueWalk, a biased random walk algorithm that samples cliques efficiently and scales quasi-linearly with the number of nodes. Together, these contributions enable the design of clique-based neural architectures that are both expressive and scalable. Extensive experiments on node and graph classification benchmarks demonstrate that our models achieve competitive or superior performance compared to GNNs and other higher-order approaches, while maintaining substantially lower memory and runtime requirements. This work establishes random walk clique-based lifting as a practical path toward scalable higher-order graph learning. It opens the door for future research on efficient sampling strategies and domain-specific applications.

REPRODUCIBILITY STATEMENT

For the developed theoretical results, we have clearly mentioned the assumptions, and complete proofs are given in Appendix B. For the experiments, we use open-source or synthetic data, and we provide a detailed description in Appendix F. For the model implementation, we provide implementation details in Appendix G, and the code will be open-sourced upon acceptance.

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

## A   Weisfeiler-Leman Graph Isomorphism Test

**Definition 21.** *Let $A(\cdot)$ and $B(\cdot)$ be graph hashing functions. We say that $A$ is more expressive than $B$ if, for any pair of graphs $G$ and $G'$, if the following condition holds:*

$$B(G) \neq B(G') \implies A(G) \neq A(G'). \tag{7}$$

Intuitively, a more expressive hashing can distinguish a wider range of non-isomorphic graphs.

A classical and widely used technique for graph isomorphism test is the *Weisfeiler–Leman (WL) test* (Leman & Weisfeiler, 1968), which is based on iterative color refinement:

**Definition 22.** *The WL test constructs, in an iterative manner, a mapping $c$ from the nodes of a graph to a finite set of colors as follows:*

1. *Initialization: All nodes are assigned the same initial color.*
2. *Color refinement: At iteration $t + 1$, the color of each node $i$ is updated according to $c_i^{t+1} = HASH\big(c_i^t, \{\{c_j^t : j \sim i\}\}\big)$, where $j \sim i$ denotes that node $j$ is adjacent to node $i$, and HASH is an injective function.*
3. *Termination: The process continues until the coloring no longer changes. Two graphs are considered non-isomorphic if their color histograms differ; otherwise, the test does not provide a conclusive answer.*

The WL test provides an efficient heuristic for the graph isomorphism problem (Huang & Villar, 2021).

## B   Proofs

### B.1   Proof of Theorem 6

First, we introduce the same notations, definitions, and propositions as in (Bodnar et al., 2021a) to manipulate cellular coloring.

**Definition 23.** *A **cellular coloring** is a function $c$ that maps a cell complex $X$ and one of its cells $\sigma$ to a finite set (color set). We denote this color as $c_\sigma^X$.*

**Definition 24.** *Let $X, Y$ be two cell complexes and $c$ a coloring. We say that $X$ and $Y$ are c-similar, denote as $c^X = c^Y$ if $\{\{c_\sigma^X, \quad \sigma \in X\}\} = \{\{c_\tau^Y, \quad \tau \in Y\}\}$. Otherwise, we have $c^X \neq c^Y$.*

**Definition 25.** *A coloring $c$ is said to **refine** another coloring $d$, denoted $c \subseteq d$, if for all cell complexes $X, Y$ and all $\sigma \in X, \tau \in Y$, we have:*

$$c_\sigma^X = c_\tau^Y \implies d_\sigma^X = d_\tau^Y.$$

*If both $c \subseteq d$ and $d \subseteq c$, then the two colorings are said to be **equivalent**, denoted $c \equiv d$.*

**Proposition 26.** *Let $X, Y$ be cell complexes with $A \subseteq X$ and $B \subseteq Y$. Consider two colorings $c, d$ such that $c \subseteq d$.*

$$\{\{c_\sigma^X, \quad \sigma \in A\}\} = \{\{c_\tau^Y, \quad \tau \in B\}\} \implies \{\{d_\sigma^X, \quad \sigma \in A\}\} = \{\{d_\tau^Y, \quad \tau \in B\}\}.$$

*Proof.* Let's suppose that $\{\{c_\sigma^X, \quad \sigma \in A\}\} = \{\{c_\tau^Y, \quad \tau \in B\}\}$. It means that there exist a bijection $f : A \to B$ such that forall $\sigma \in A$, $c_\sigma^X = c_{f(\sigma)}^Y$.
As $c \subseteq d$, $d_\sigma^X = d_{f(\sigma)}^Y$ ie $\{\{d_\sigma^X, \quad \sigma \in A\}\} = \{\{d_\tau^Y, \quad \tau \in B\}\}$. $\qquad\square$

**Corollary 27.** *If $c \subseteq d$, then for all cell complexes $X, Y$,*

$$c^X = c^Y \implies d^X = d^Y.$$

*All non-distinguished cell complexes by $c$ are not distinguished by $d$. In other words, $c$ is a more powerful isomorphic test than $d$.*

**Proof of Theorem 6.** Let's show that CWL with coloring $\mathrm{HASH}(c_\sigma^t, c_\mathcal{B}^t, c_\mathcal{C}^t)$ is as powerful as $\mathrm{HASH}(c_\sigma^t, c_\mathcal{B}^t, c_\uparrow^t)$. Let's denote as $a^t$ the colouring at step $t$ using CWL with $\mathrm{HASH}(c_\sigma^t, c_\mathcal{B}^t, c_\uparrow^t)$ and $b^t$ the one using $\mathrm{HASH}(c_\sigma^t, c_\mathcal{B}^t, c_\mathcal{C}^t)$. We know that the coloring $a^t$ is as powerful as the original CWL (Theorem 7, in Bodnar et al. (2021a)). Since $b^t$ uses a subset of the CWL coloring relationships, it can be shown by induction that it is less powerful than the original CWL. Therefore, we have $a \sqsubseteq b$.

Let's show that $b \sqsubseteq a$.

We show by induction that $b^{2t} \sqsubseteq a^t$ for all $t \in \mathbb{N}$.

*Base case.* $b^0 \sqsubseteq a^0$ as they follow the same color initialization scheme.

*Inductive step.* Assume $b^{2t} \sqsubseteq a^t$. We prove that $b^{2t+2} \sqsubseteq a^{t+1}$.

let $(\sigma_1, \sigma_2) \in X \times Y$ such that $b_{\sigma_1}^{2t+2} = b_{\sigma_2}^{2t+2}$. By construction,

$$b_{\sigma_1}^{2t+1} = b_{\sigma_2}^{2t+1}, \quad b_\mathcal{B}^{2t+1}(\sigma_1) = b_\mathcal{B}^{2t+1}(\sigma_2), \quad b_\mathcal{C}^{2t+1}(\sigma_1) = b_\mathcal{C}^{2t+1}(\sigma_2),$$

as $b_\mathcal{C}^{2t+1}(\sigma_1) = b_\mathcal{C}^{2t+1}(\sigma_2)$, there exist a bijective map $f : \mathcal{C}(\sigma_1) \to \mathcal{C}(\sigma_2)$ that preserve the $b^{2t+1}$ coloring ie $b_\tau^{2t+1} = b_{f(\tau)}^{2t+1}$ for $\tau \in \mathcal{C}(\sigma_1)$.

As $b_\tau^{2t+1} = b_{f(\tau)}^{2t+1}$, we have $b_\mathcal{B}^{2t}(\tau) = b_\mathcal{B}^{2t}(f(\tau))$, *i.e.*,

$$\{\!\{b_\gamma^{2t}, \quad \gamma \in \mathcal{B}(\tau)\}\!\} = \{\!\{b_\gamma^{2t}, \quad \gamma \in \mathcal{B}(f(\tau))\}\!\}.$$

We can add the color of $\tau$ on both sides, the multisets would still stay equal:

$$\{\!\{(b_\gamma^{2t}, b_\tau^{2t}), \quad \gamma \in \mathcal{B}(\tau)\}\!\} = \{\!\{(b_\gamma^{2t}, b_\tau^{2t}), \quad \gamma \in \mathcal{B}(f(\tau))\}\!\}.$$

As this is true for all $\tau$ in $\mathcal{C}(\sigma_1)$, we can take the union:

$$\bigcup_{\tau \in \mathcal{C}(\sigma_1)} \{\!\{(b_\gamma^{2t}, b_\tau^{2t}), \quad \gamma \in \mathcal{B}(\tau)\}\!\} = \bigcup_{\tau \in \mathcal{C}(\sigma_1)} \{\!\{(b_\gamma^{2t}, b_\tau^{2t}), \quad \gamma \in \mathcal{B}(f(\tau))\}\!\},$$

*i.e.*,

$$\{\!\{(b_\gamma^{2t}, b_\tau^{2t}), \quad \tau \in \mathcal{C}(\sigma_1), \gamma \in \mathcal{B}(\tau)\}\!\} = \{\!\{(b_\gamma^{2t}, b_\tau^{2t}), \quad \tau \in \mathcal{C}(\sigma_1), \gamma \in \mathcal{B}(f(\tau))\}\!\},$$

as $b_\tau^{2t} = b_{f(\tau)}^{2t}$ and $f$ is bijective, the right term can be simplified:

$$\{\!\{(b_\gamma^{2t}, b_\tau^{2t}), \quad \tau \in \mathcal{C}(\sigma_1), \gamma \in \mathcal{B}(f(\tau))\}\!\} = \{\!\{(b_\gamma^{2t}, b_{f(\tau)}^{2t}), \quad \tau \in \mathcal{C}(\sigma_1), \gamma \in \mathcal{B}(f(\tau))\}\!\}$$
$$= \{\!\{(b_\gamma^{2t}, b_\delta^{2t}), \quad \delta \in \mathcal{C}(\sigma_2), \gamma \in \mathcal{B}(\delta)\}\!\},$$

*i.e.*,

$$\{\!\{(b_\gamma^{2t}, b_\tau^{2t}), \quad \tau \in \mathcal{C}(\sigma_1), \gamma \in \mathcal{B}(\tau)\}\!\} = \{\!\{(b_\gamma^{2t}, b_\delta^{2t}), \quad \delta \in \mathcal{C}(\sigma_2), \gamma \in \mathcal{B}(\delta)\}\!\}.$$

Thus $b_\uparrow^{2t}(\sigma_1) = b_\uparrow^{2t}(\sigma_2)$. Using the induction hypothesis $b^{2t} \sqsubseteq a^t$ with proposition 26, we have

$$a_{\sigma_1}^t = a_{\sigma_2}^t \quad a_\uparrow^t(\sigma_1) = a_\uparrow^t(\sigma_2) \quad a_\mathcal{B}^t(\sigma_1) = a_\mathcal{B}^t(\sigma_2) \quad a_\mathcal{C}^t(\sigma_1) = a_\mathcal{C}^t(\sigma_2),$$

*i.e.*,

$$a_{\sigma_1}^{t+1} = a_{\sigma_2}^{t+1}.$$

From our induction $b^{2t} \sqsubseteq a^t$ for all $t \in \mathbb{N}$, hence $b \sqsubseteq a$. $\qquad\square$

## B.2 PROOF OF THEOREM 8 AND PROPOSITION 12

We introduce a new isomorphism test, fCWL, associated with fCWN, and prove that fCWL is at least as expressive as CWL and 1-WL on cell complexes that kept node set.

Once this is established, the remaining correspondences between models with injective aggregation and their associated tests follow identically from the proof of equivalence between CWL and CWN in (Bodnar et al., 2021a).

**Proposition 28.** *fCWL is more expressive than sCWL.*

*Proof.* $(\mathcal{V}_1, \mathcal{X}_1)$ and $(\mathcal{V}_2, \mathcal{X}_2)$ correspond to two cell complexes that keep node sets.

Let $a^t$ denote the coloring at step $t$ using sCWL, and $b^t$ the coloring at step $t$ using fCWL.

We prove by induction that $b^t \subseteq a^t$.

*Base case.* $b^0 \subseteq a^0$ since both follow the same initialization scheme.

*Induction step.* Assume $b^t \subseteq a^t$. We show that $b^{t+1} \subseteq a^{t+1}$.

Let $(\sigma_1, \sigma_2) \in \mathcal{X}_1 \times \mathcal{X}_2$ such that $b_{\sigma_1}^{t+1} = b_{\sigma_2}^{t+1}$. By construction, we have:

$$b_{\sigma_1}^t = b_{\sigma_2}^t, \quad b_{\mathcal{B}}^t(\sigma_1) = b_{\mathcal{B}}^t(\sigma_2), \quad b_{\mathcal{C}}^t(\sigma_1) = b_{\mathcal{C}}^t(\sigma_2).$$

Using Proposition 26 with the induction hypothesis, it follows that:

$$a_{\sigma_1}^t = a_{\sigma_2}^t, \quad a_{\mathcal{B}}^t(\sigma_1) = a_{\mathcal{B}}^t(\sigma_2), \quad a_{\mathcal{C}}^t(\sigma_1) = a_{\mathcal{C}}^t(\sigma_2)$$

*i.e.,* $a_{\sigma_1}^{t+1} = a_{\sigma_2}^{t+1}$.

By induction, $b^t \subseteq a^t$ for all $t \in \mathbb{N}$, hence $b \subseteq a$. □

Since sCWL is as expressive as CWL (Theorem 6), it follows as a corollary that fCWL is at least as expressive than CWL.

**Proposition 29.** *fCWL is more expressive than WL*

*Proof.* $(\mathcal{V}_1, \mathcal{X}_1)$ and $(\mathcal{V}_2, \mathcal{X}_2)$ correspond to two cell complexes that keep node sets.

Let $a^t$ denote the coloring of nodes at step $t$ using WL, $b^t$ the coloring of cells at step $t$ using fCWL and $b_{\mathcal{V}}^t$ the coloring of nodes in the cell complex colored at step $t$ by fCWL.

We prove by induction that $b_{\mathcal{V}}^t \subseteq a^t$ on the nodes.

*Base case.* $b^0 \subseteq a^0$ since have constant colors.

*Induction step.* Assume $b_{\mathcal{V}}^t \subseteq a^t$ on nodes. We show that $b_{\mathcal{V}}^{t+1} \subseteq a^{t+1}$.

Let $(i_1, i_2) \in \mathcal{V}_1 \times \mathcal{V}_2$ such that $b_{i_1}^{t+1} = b_{i_2}^{t+1}$.

We have:

$$b_{i_1}^t = b_{i_2}^t, \quad b_{\mathcal{C}(i_1)}^t = b_{\mathcal{C}(i_2)}^t, \quad b_{\mathcal{N}(i_1)}^t = b_{\mathcal{N}(i_2)}^t.$$

Using the induction hypothesis: $a_{i_1}^t = a_{i_2}^t$. as $b_{\mathcal{N}(i_1)}^t = b_{\mathcal{N}(i_2)}^t$, we can only consider the color of the first component, we get:

$$\{\{b_j^t, \quad j \in \mathcal{N}(i_1)\}\} = \{\{b_j^t, \quad j \in \mathcal{N}(i_2)\}\},$$

*i.e.,* by using proposition 26 and the induction hypothesis:

$$\{\{a_j^t, \quad j \in \mathcal{N}(i_1)\}\} = \{\{a_j^t, \quad j \in \mathcal{N}(i_2)\}\}.$$

From WL update, we get $a_{i_1}^{t+1} = a_{i_2}^{t+1}$.

By induction. $b_{\mathcal{V}}^t \subseteq a^t$ for all $t \in \mathbb{N}$, thus $b_{\mathcal{V}} \subseteq a$. □

## B.3 PROOF OF PROPOSITION 14

In this section, we analyse the theoretical time and memory complexity of CWN, fCWN, and sCWN. We first remind some notations:

- $\mathcal{V}$ represents the set of nodes
- $n$ is the number of nodes of our graphs
- $\mathcal{N}_i$ represents the neighborhood of node $i$.
- $\mathcal{X}$ is the set of maximal cliques.

We now detail one by one each message passing scheme's complexity.

**Boundary messages.** Each node in the graph sends a message to the clique containing it. The total number of messages sent is:

$$|\{(i,\sigma) \in \mathcal{V} \times \mathcal{X}, \quad i \in \sigma\}| = \sum_{(i,\sigma) \in \mathcal{V} \times \mathcal{X}} \mathbb{1}_{i \in \sigma} = \sum_{\sigma \in \mathcal{X}} \sum_{i \in \mathcal{V}} \mathbb{1}_{i \in \sigma} = \sum_{\sigma \in \mathcal{X}} |\sigma|.$$

**Co-boundary messages.** Each clique sends a message to each node it contains. The total number of messages sent is:

$$|\{(i,\sigma) \in \mathcal{V} \times \mathcal{X}, \quad i \in \sigma\}| = \sum_{\sigma \in \mathcal{X}} |\sigma|.$$

**Upper-adjacency CWN.** Each node $i$ aggregate message for all tuple $(j,\sigma)$ such that $\{i,j\} \subset \sigma$. The total number of messages sent is:

$$\sum_{i \in \mathcal{V}} \left|\{(j,\sigma) \in \mathcal{V} \times \mathcal{X} : \{i,j\} \in \sigma\}\right| = \sum_{i \in \mathcal{V}} \sum_{j \in \mathcal{V}} \sum_{\sigma \in \mathcal{X}} \mathbb{1}_{\{i,j\} \subset \sigma}$$

$$= \sum_{\sigma \in \mathcal{X}} \sum_{i \in \mathcal{V}} \sum_{j \in \mathcal{V}} \mathbb{1}_{\{i,j\} \subset \sigma}$$

$$= \sum_{\sigma \in \mathcal{X}} \left|\{(i,j) \in \mathcal{V}^2 : \{i,j\} \subset \sigma\}\right|$$

$$= \sum_{\sigma \in \mathcal{X}} \binom{|\sigma|}{2}$$

$$= \sum_{\sigma \in \mathcal{X}} \frac{|\sigma|^2 - |\sigma|}{2}.$$

**Upper-adjacency fCWN.** For each tuple $(i,\sigma) \in \mathcal{V} \times \mathcal{X}$ we create a message. Then we do an adjacency update. The total number of messages is the sum of each:

$$\sum_{(i,\sigma) \in \mathcal{V} \times \mathcal{X}} \mathbb{1}_{i \in \sigma} + \sum_{l \in \mathcal{N}_i} 1 = \sum_{\sigma \in \mathcal{X}} |\sigma| + |\mathcal{E}|.$$

We can now finish the proof of proposition 14.

**CWN.** Every message passes through an MLP $M_\uparrow$. The memory complexity is the same as the number of messages plus the data on the node and cliques:

- Time complexity : $\mathcal{O}(\sum_{\sigma \in \mathcal{X}} |\sigma|^2)$.
- Memory complexity : $\mathcal{O}(n + \sum_{\sigma \in \mathcal{X}} |\sigma|^2)$.

**fCWN.** Only the first messages go through an MLP $M_\uparrow$.

- Time complexity : $\mathcal{O}(\sum_{\sigma \in \mathcal{X}} |\sigma| + |\mathcal{E}|)$.
- Memory complexity : $\mathcal{O}(n + \sum_{\sigma \in \mathcal{X}} |\sigma|)$.

**sCWN.** Here, MLPs are only applied to node or clique data. The messages are based on boundary and co-Boundary.

- Time complexity : $\mathcal{O}(\sum_{\sigma \in \mathcal{X}} |\sigma|)$.
- Memory complexity : $\mathcal{O}(n + |\mathcal{X}|)$.

**Summary.** For clarity, we summarize below:

| Model | Time Complexity | Memory Complexity |
|---|---|---|
| CWN | $\mathcal{O}(\sum_{\sigma \in \mathcal{X}} \lvert \sigma \rvert^2)$ | $\mathcal{O}(n + \sum_{\sigma \in \mathcal{X}} \lvert \sigma \rvert^2)$ |
| fCWN | $\mathcal{O}(\sum_{\sigma \in \mathcal{X}} \lvert \sigma \rvert + \lvert \mathcal{E} \rvert)$ | $\mathcal{O}(n + \sum_{\sigma \in \mathcal{X}} \lvert \sigma \rvert)$ |
| sCWN | $\mathcal{O}(\sum_{\sigma \in \mathcal{X}} \lvert \sigma \rvert)$ | $\mathcal{O}(n + \lvert \mathcal{X} \rvert)$ |

## B.4 PROOF OF PROPOSITION 18

We show that at every step of Algorithm 1, the nodes in the walk always form a clique.

**Notations.** Let $\text{Walk}_t$ denote the nodes in the walk at step $t$, and $\text{neighbor}_t$ the set of nodes that can be added next. We claim that:

$$\text{neighbor}_t = \{l \in \mathcal{V}, \quad l \sim j \ \forall j \in \text{Walk}_t\},$$

*i.e.*, $\text{neighbor}_t$ contains exactly the nodes connected to all nodes in the current walk.

**Induction.**

*Base case.* Initially, $\text{Walk}_0 = [i]$ and $\text{neighbor}_0 = \mathcal{N}_i$. By definition, $\mathcal{N}_i$ contains all nodes connected to $i$, i.e., all nodes that form a clique with $\text{Walk}_0$. Thus, the property holds at the first step.

*Inductive step.* Assume the property holds at step $t$, and let $j_{\text{new}} \in \text{neighbor}_t$ be the next node added to the walk. The neighbor set is updated as

$$\text{neighbor}_{t+1} = \text{neighbor}_t \cap \mathcal{N}_{j_{\text{new}}}.$$

By construction, $\text{neighbor}_{t+1}$ contains only nodes connected to $j_{\text{new}}$ and to all nodes in $\text{Walk}_t$, i.e., nodes connected to all nodes in

$$\text{Walk}_{t+1} = \text{Walk}_t \cup \{j_{\text{new}}\}.$$

The property holds at step $t + 1$.

**Conclusion.** By induction, all nodes in the walk are connected to each other, *i.e.*, the walk always forms a clique. Since the walk is a clique, its size cannot exceed $\omega(G)$, the size of the largest clique in the graph. Therefore, the walk can only stop when $\text{neighbor}_t$ becomes empty, *i.e.*, when there is no node that can be added to extend the clique. As a result, the clique produced by the walk is maximal with respect to set inclusion.

## B.5 PROOF OF PROPOSITION 19

CliqueWalk builds a maximal clique by growing it step by step. At each step, the algorithm: (i) samples a neighbor, (ii) intersects the neighborhoods of the current and newly visited node to restrict the walk, and (iii) continues until either the walk length reaches $\omega_{\text{max}}$ or it cannot be expanded.

We can now break down the cost of one walk:

(i) *Neighbor sampling.* Selecting a random neighbor is constant-time: $O(1)$.
(ii) *Neighborhood intersection.* Intersecting two neighborhoods $A$ and $B$ takes $O(\lvert A \rvert + \lvert B \rvert)$. Since each neighborhood is bounded by the maximum degree $d_{\text{max}}(G)$, this step costs at most $O(d_{\text{max}}(G))$.
(iii) *Walk length.* The maximum length of a walk is bounded by

$$L \leqslant \max\big(\omega(G), \omega_{\text{max}}\big),$$

where $\omega(G)$ is the maximum clique size of the graph and $\omega_{\text{max}}$ is the cutoff imposed by the algorithm.

Table 4: Number of distinct hashes found by each method on graph classification datasets. Abbreviated dataset names: ENZ = ENZYMES, FRANK = FRANKENSTEIN, IMDB-B = IMDB-BINARY, IMDB-M = IMDB-MULTI, PROT = PROTEINS, ALC = alchemy_full.

| Method | DD | ENZ | FRANK | IMDB-B | IMDB-M | NCI1 | PROT | ALC |
|--------|------|-----|-------|--------|--------|------|------|-------|
| 1WL | 1178 | 595 | 2766 | 537 | 387 | 3837 | 996 | 12343 |
| CWL | 1178 | 595 | 2767 | 537 | 387 | 3837 | 996 | 12396 |
| CountClique | 1178 | 547 | 216 | 432 | 309 | 254 | 799 | 23 |
| TopoCount | 1178 | 595 | 1272 | 537 | 387 | 2188 | 992 | 727 |

The complexity of one CLiqueWalk is thus :

$$\mathcal{O}(\sum_{j=0}^{L} d_{\max(G)}) = \mathcal{O}\left(d_{\max}(G) \cdot \max(\omega(G), \omega_{\max})\right).$$

As we launch from each node $n_{\text{walks}}$ walks, the total complexity is

$$O(n \cdot n_{\text{walks}} \cdot d_{\max}(G) \cdot \max(\omega(G), \omega_{\max})). \quad \square$$

## C MAXIMAL CLIQUE CWL

We propose some experiments and illustrations to better understand the maximal clique CWL and its differences with WL. See Figure 6. It is known that CWL is more expressive than WL when using cell lifting methods that preserve the full node and edge sets of the graph (Bodnar et al., 2021a). However, since we only consider maximal cliques and remove edges from the representation, we no longer have this guarantee over WL.

We introduce two simple coloring scheme to make sense of CWL expressive power.

**Definition 30.** *The **CountClique** test hashes the set of all clique lengths.*

**Definition 31.** *The **TopoCount** test assigns a unique color to each node by hashing the set of lengths of the cliques containing it.*

It is clear that CWL is at least as expressive as TopoCount and CountClique.

We empirically compare the expressivity of CWL, WL, and other tests on various datasets. Table 4 shows the number of distinct hashes produced by each method. CWL matches or slightly exceeds WL in most cases. For several datasets (Dobson & Doig, 2003b; Chen et al., 2019; Orsini et al., 2015), access to clique neighborhood information allows CWL to distinguish more graphs. For chemical datasets such as *alchemy_full*, WL schemes produce significantly more hashes than one-shot methods like TopoCount, highlighting the benefit of multi-layer models on those datasets.

We also evaluate these tests on strongly regular graphs (see Figure 7a and Table 5). We use strongly regular datasets from https://www.maths.gla.ac.uk/~es/srgraphs.php (Haemers & Spence, 2001), which include non-isomorphic strongly regular graphs with up to 64 nodes. For many strongly regular graph families, clique topology alone is sufficient to distinguish most graphs. In contrast, 1WL and 3WL fail to discriminate any graphs in these families, which aligns with known results (Bouritsas et al., 2022; Bodnar et al., 2021a).

**Clique against cycle lifting.** Figure 7b compares CWL with node and maximal clique lifting against CWL with node, edge, and cycle lifting. Both approaches achieve similar graph discriminative power, though they are not directly comparable: in some cases, cliques distinguish more graphs, while in others, cycles do.

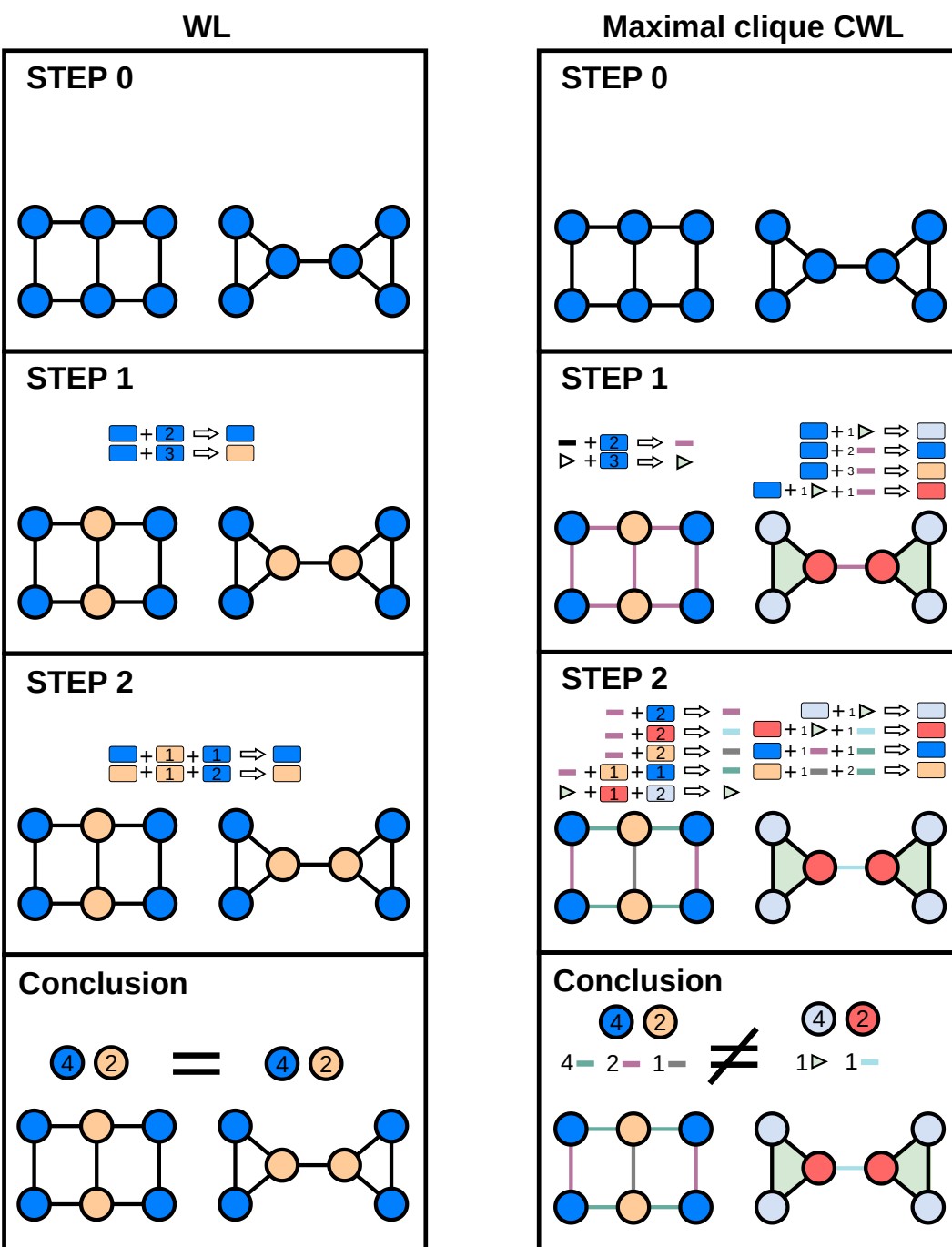

Figure 6: Illustration of the WL and maximal clique CWL test. At each iteration, every node updates its color based on its own color and the colors of its neighboring structures (see Steps 1 and 2). After Step 2, the colors become stable (*i.e.*, invariant under further updates), and the algorithm stops. A histogram of colors is then computed. Since the two graphs produce identical histograms for WL, the test cannot distinguish between them, and the WL test is therefore inconclusive. In contrast, the maximal-clique CWL algorithm yields different histograms for the two graphs, allowing us to conclude that they are not isomorphic.

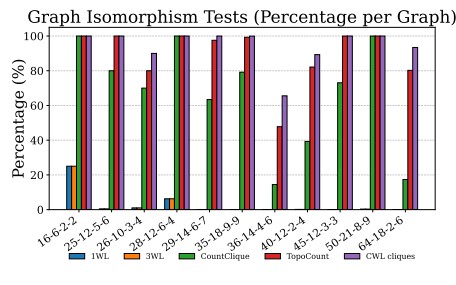
(a) Clique CWL against other tests

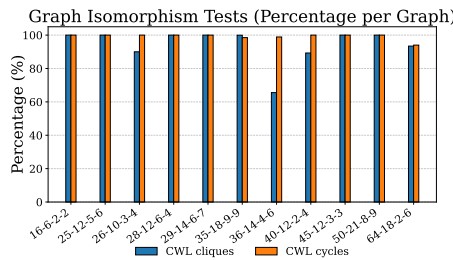
(b) Clique CWL against cycle CWL

Figure 7: Comparison of Percentage of Unique Graph Hashes on strongly regular datasets: (a) compare CWL on maximal cliques against other isomorphic tests, (b) compare CWL on maximal cliques against CWL on node, edge, cycle lifting.

Table 5: Number of graphs in each strongly regular family.

| Family | Number of graphs |
| --- | --- |
| 16-6-2-2 | 2 |
| 25-12-5-6 | 15 |
| 26-10-3-4 | 10 |
| 28-12-6-4 | 4 |
| 29-14-6-7 | 41 |
| 35-18-9-9 | 3854 |

| Family | Number of graphs |
| --- | --- |
| 36-14-4-6 | 180 |
| 40-12-2-4 | 28 |
| 45-12-3-3 | 78 |
| 50-21-8-9 | 18 |
| 64-18-2-6 | 167 |

# D CLIQUE SAMPLING

A classical approach for enumerating all maximal cliques is the *Bron-Kerbosch* method (Bron & Kerbosch, 1973), explained in Algorithm 2. $R$ is the current clique being grown, $P$ contains nodes adjacent to all vertices in $R$, and $X$ contains nodes already processed that are also adjacent to every vertex in $R$. Clique summarization has been widely studied (D'Elia et al., 2025). Most of those approaches modify the Brun-Kerbosch algorithm to enumerate or sample a subset of the maximal clique set that verifies specific properties. For instance, Wang et al. (2013) prunes branches based on a heuristic to construct a subset of maximal cliques that partially covers all maximal cliques.

---
**Algorithm 2** Bron–Kerbosch

1: **procedure** BRONKERBOSCH($R, P, X$)
2:  **if** $P = \varnothing$ and $X = \varnothing$ **then**
3:    report $R$ as a maximal clique
4:  **else**
5:    **for** each $u$ in a copy of $P$ **do**
6:      $P \leftarrow P \backslash \{u\}$
7:      $R_{\text{new}} \leftarrow R \cup \{u\}$
8:      $P_{\text{new}} \leftarrow P \cap N(u)$
9:      $X_{\text{new}} \leftarrow X \cap N(u)$
10:     BRONKERBOSCH($R_{\text{new}}, P_{\text{new}}, X_{\text{new}}$)
11:     $X \leftarrow X \cup \{u\}$
12:   **end for**
13: **end if**
14: **end procedure**

---

Our method, *CliqueWalk*, is also inspired by Bron-Kerbosch but differs in two important ways: (*i*) *We sample rather than full enumeration.* CliqueWalk does not attempt to enumerate all maximal cliques but samples a subset of them. Therefore, (*ii*) *we do not need the X set.* We simply grow a clique by iteratively sampling a vertex from the candidate set $P$. Conceptually, CliqueWalk performs an upward random walk in the clique complex (see Figure 3). While exact clique sampling might require exploring a geometric number of recursive branches (see Proposition 17), CliqueWalk runs in linear time with respect to the number of nodes (see Proposition 19) and efficiently produces summaries of the clique topology with the following sampling guarantees: (*i*) The sampling process tends to sample larger cliques. For instance, given a node $v$ and a maximal clique $\sigma$ containing $v$, the probability of sampling $\sigma$ is at most $(|\sigma| - 1)/deg(v)$. (*ii*) Performing CliqueWalk with multiple walks per node ensures that each node is included in several sampled cliques, which is relevant for node-level learning tasks.

## E  ABLATIONS

**Cell input feature choice.**  Table 6 compares the performance of sCWN on *Photo* and *contact-primary-school* depending on the type of input used. We observe that size embedding and sum embedding obtain very similar accuracy, whereas mean embedding provides much worse results on contact-high-school.

Table 6: Ablation cell input features. Table report test accuracy after training.

| input type
dataset | size emb | sum | mean |
|---|---|---|---|
| Photo | 94.7% | 94.5% | 94.9% |
| contact-high-school | 95.4% | 97.6% | 7.0% |

**Number of layers effects.**  Figure 8 shows the evolution of the accuracy for deeper models. As depth increases, test accuracy degrades at some point, indicating that deep models struggle to learn effectively. Training and testing accuracy remain similar at large depths (not shown in the figure), this decline is unlikely due to over-fitting and is consistent with the over-smoothing effect known in graph learning Einizade et al. (2025).

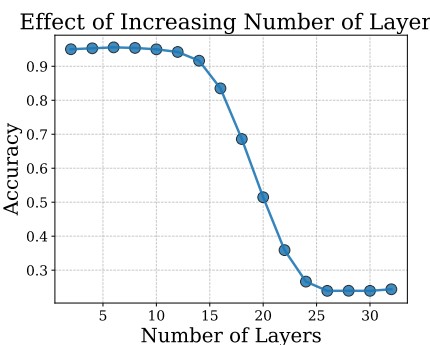

Figure 8:  Accuracy of trained sCWN model without batchnorm on Photo depending on the number of layers.

**Sampling effects.**  As in Section 5.4, we compare exact enumeration of maximal cliques with CliqueWalk sampling using between 1 and 256 walks per node on *contact-primary-school* (Figure 9). For sCWN, performance is better with fewer sampled structures, suggesting that excessive redundancy may dilute useful information, especially for a large number of walks, where the number of sampled maximal cliques can exceed the number of nodes by a large margin. In contrast, fCWN remains relatively stable across different sampling rates, indicating that its message-passing scheme is more robust across different sampling rates.

## F  DATASETS

**Topological networks** (Chodrow et al., 2021; Mastrandrea et al., 2015). The *contact-high-school* and *contact-primary-school* datasets record proximity between students. Hyperedges are created at fixed time intervals from these interactions. We then project all interactions into a static graph. In this graph, an edge links two students if they have interacted at least once. The resulting graphs are topological complex networks (See Figures 10a and 10b)

**Citation networks.**  In these datasets, node features are given by a Bag-of-Words representation of the documents. Cora and Citeseer are citation networks extracted from machine learning publications (Sen et al., 2008). The labels correspond to the research topic of each paper. The PubMed citation network consists of articles related to diabetes. (Namata et al., 2012) The labels indicate the type of diabetes discussed in the article.

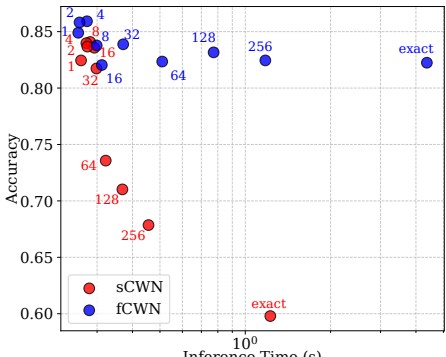

Figure 9:  Accuracy of CWN and SCCN at different CliqueWalk sampling rates on contact-primary-school.

**Purchase network.**  The Amazon Photo dataset is a subset of the Amazon co-purchase network (McAuley et al., 2015). In this graph, nodes represent products, and edges connect items that are frequently purchased together. node features are given by a Bag-of-Words representation of product reviews, and the labels are the product category. The OGBN-Products dataset follows the same methodology, but the Bag-of-Words features have been reduced to 100 dimensions using PCA, providing a more compact representation of the node features.

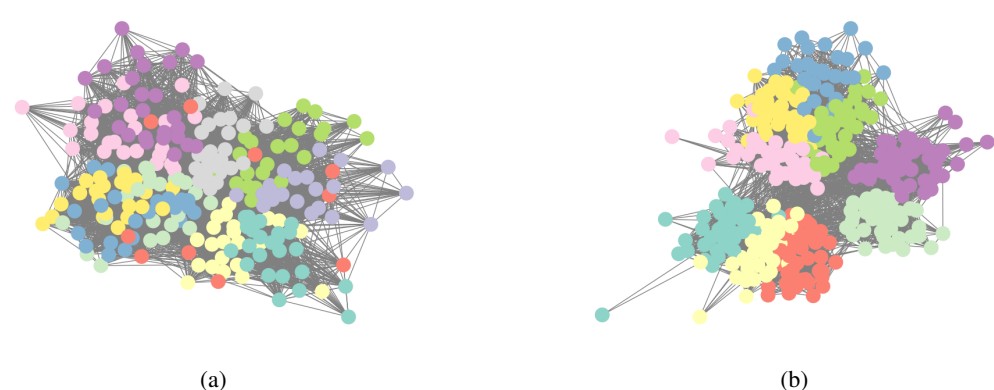

Figure 10: Projected datasets: (a) contact-primary-school and (b) contact-high-school.

**Stochastic clique model.** It is a special case of Stochastic Block Model (Holland et al., 1983) where inward probability is set to 1. Graphs are generated by assembling cliques, with nodes inside each clique fully connected. Each clique is assigned a label, which is inherited by all its nodes, and node features are generated from a Gaussian distribution with a mean determined by the node label and a fixed diagonal variance. To introduce topological noise, each node is connected to nodes outside its clique with a fixed probability, perturbing the clique structure. The task can thus be interpreted as a form of label denoising. For our experiments reported in table For experiments reported in Table 1, cliques had random sizes between 10 and 20. Node features had a standard deviation of 2, and topological noise was such that approximately two out of three neighbors came from outside the clique. Each clique was assigned one of five possible labels.

**Social networks.** A network of actors and actresses is constructed from IMDB, where edges indicate collaboration in the same film. The *IMDB-BINARY* and *IMDB-MULTI* datasets (Yanardag & Vishwanathan, 2015) consist of the 1-hop neighborhoods around each actor. Graph labels correspond to the movie genre associated with the actor.

**Bioinformatics.** The bioinformatics datasets include four widely used molecular and protein graph collections. *MUTAG* (Debnath et al., 1991) contains nitroaromatic compounds with 7 different labels indicating mutagenic activity. *PROTEINS* (Borgwardt et al., 2005) represents protein structures; the task is to predict if a protein is an enzyme or not. *NCI1* and *NCI109* (Wale et al., 2008; Shervashidze et al., 2010) are collections of chemical compounds tested for activity against lung cancer and ovarian cancer cells, respectively. Each dataset is available through the TUDataset (Morris et al., 2020) repository and is commonly used to benchmark graph-based learning methods.

*Remark.* Dataset statistics can be found in Table 7. Clique size where approximated using Clique-Walk for OGBN-Products.

**OBGN-Products.** We used fixed hyperparameters for all models: a learning rate of $10^{-3}$, no dropout, a hidden dimension of 64, and 3 layers with batch normalisation. The experimental setup was kept intentionally simple, without node batching. For higher accuracy, we recommend using larger hidden dimensions, deeper architectures, and node batching, as models with more parameters and efficient training generally perform better on large datasets.

## G  MODEL AND LAYER DETAILS

In this section, we describe the layers and model implementations used for our benchmarks.

Throughout, we use the following notation:

- MLP: a 2-layer multilayer perceptron with ReLU activation.
- **W**: a learnable linear layer.
- $\mathbf{H} \in \{0, 1\}^{n \times m}$: the hypergraph incidence matrix.

Table 7: Dataset statistics for node and graph classification. Reported are the number of nodes, number of edges, mean degree, and clique statistics ($\mu$: mean size, $\sigma$: standard deviation).

| Dataset | Nodes | Edges | Mean degree | Clique $\mu$ | Clique $\sigma$ |
|---------|-------|-------|-------------|--------------|-----------------|
| *Node classification datasets* | | | | | |
| SCM | 6 002 010 | 276 089 116 | 46.0 | 6.51 | 6.55 |
| Cora | 2 708 | 10 556 | 7.80 | 2.37 | 0.59 |
| PubMed | 19 717 | 88 648 | 8.99 | 2.28 | 0.59 |
| Citeseer | 3 327 | 9 104 | 5.47 | 2.26 | 0.58 |
| Photo | 7 650 | 238 162 | 62.26 | 10.75 | 4.89 |
| Contact-Primary-School | 242 | 16 634 | 137.47 | 11.36 | 2.88 |
| Contact-High-School | 327 | 11 636 | 71.17 | 9.28 | 3.73 |
| OGBN-Products | 2 449 029 | 123 718 024 | 50.5 | 8.3 | 6.6 |
| *Graph classification datasets* | | | | | |
| IMDB-BINARY | 19 773 | 96 531 | 9.76 | 7.02 | 3.80 |
| IMDB-MULTI | 19 502 | 98 903 | 10.14 | 7.61 | 4.30 |
| MUTAG | 3 371 | 3 721 | 2.21 | 2.00 | 0.00 |
| NCI1 | 122 747 | 132 753 | 2.16 | 2.00 | 0.04 |
| NCI109 | 122 494 | 132 604 | 2.17 | 2.00 | 0.04 |
| Proteins | 43 471 | 81 044 | 3.73 | 2.53 | 0.63 |

- $\mathbf{D}_v \in \mathbb{R}^{n \times n}$, $\mathbf{D}_e \in \mathbb{R}^{m \times m}$: diagonal degree matrices of nodes and hyperedges (cliques):

$$\mathbf{D}_v(i,i) = \sum_{e=1}^{m} \mathbf{H}_{i,e}, \quad \mathbf{D}_e(e,e) = \sum_{i=1}^{n} \mathbf{H}_{i,e}.$$

- $\mathcal{X}$: the set of cliques.
- $\mathbf{x}_i^N$: features of node $i \in \mathcal{V}$.
- $\mathbf{x}_\sigma^C$: features of clique $\sigma \in \mathcal{X}$.

**HGNN.** We follow (Feng et al., 2019). The layer propagation is:

$$\mathbf{x}_i^N \leftarrow \mathbf{W}\mathbf{x}_i^N + \mathbf{W}\mathbf{D}_v^{-\frac{1}{2}}\mathbf{H}\mathbf{D}_e^{-1}\mathbf{H}^\top \mathbf{D}_v^{-\frac{1}{2}}\mathbf{W}(\mathbf{x}_i^N),$$

where $\mathbf{W}$ is a learnable weight matrix, and $\sigma(\cdot)$ is a non-linear activation function (*e.g.*, ReLU). The addition of $\mathbf{X}^{(l)}$ implements a residual (skip) connection.

**CWN.** We implemented the layer from Bodnar et al. (2021a):

$$\mathbf{x}_\sigma^C \leftarrow \mathrm{MLP}\Big(\mathbf{x}_\sigma^C + \frac{1}{|\sigma|}\sum_{i \in \sigma} \mathbf{x}_i^N\Big),$$

$$\mathbf{x}_i^N \leftarrow \mathbf{W}\mathbf{x}_i^N + \frac{1}{|\{(j,\sigma) : i, j \in \sigma\}|} \sum_{\substack{(j,\sigma) \\ i,j \in \sigma}} \mathrm{MLP}\big(\mathbf{x}_i^N + \mathbf{x}_j^N + \mathbf{x}_\sigma^C\big).$$

**fCWN.** We implemented the layer:

$$\mathbf{x}_\sigma^C \leftarrow \frac{1}{|\sigma|}\sum_{i \in \sigma} \mathbf{x}_i^N,$$

$$\mathbf{m}_i \leftarrow \frac{1}{|\{\sigma : \sigma \ni j\}|} \sum_{\sigma \ni j} \mathrm{MLP}\big(\mathbf{x}_j^N + \mathbf{x}_\sigma^C\big)$$

$$\mathbf{x}_i^N \leftarrow \mathbf{W}\mathbf{x}_i^N + \mathbf{W}\mathbf{m}_i + \frac{1}{|\mathcal{N}_i|}\sum_{j \in \mathcal{N}_i} \mathbf{m}_j.$$

**sCWN.** This model is a simple boundary, co-boundary aggregation. Most of the weights are used to update clique representation, while node representations are updated from the average of clique features.

$$\mathbf{x}_\sigma^C \leftarrow \mathrm{MLP}\Big(\mathbf{W}\mathbf{x}_\sigma^C + \frac{1}{|\sigma|}\sum_{i\in\sigma}\mathrm{MLP}(\mathbf{x}_i^N)\Big),$$

$$\mathbf{x}_i^N \leftarrow \mathbf{W}\mathbf{x}_i^N + \frac{1}{|\{\sigma\in\mathcal{X}:i\in\sigma\}|}\sum_{\sigma\ni i}\mathbf{x}_\sigma^C.$$

**SCCN.** We used the TopoModelX (Hajij et al., 2024) implementation of the SCCN layer from (Yang et al., 2022).

**Global architecture.** ach model begins with a layer normalization of the input. Each subsequent layer is composed as follows:

$$\mathrm{Conv} \rightarrow \mathrm{ReLU} \rightarrow \mathrm{BatchNorm}\ (\text{with or without}) \rightarrow \mathrm{Dropout}.$$

Where Conv can be replaced with any convolutional layer under evaluation (*e.g.* sCWN, SCCN, GAT, etc.).

**Graph models.** We experiment with several standard graph neural networks: Simple Graph Convolution (SGC), Graph Convolutional Network (GCN), GraphSAGE, Graph Attention Network (GAT), and Graph Isomorphism Network (GIN). For SGC, we use a modified version with shift operator $\mathbf{S} := \mathbf{D}^{-1}\mathbf{A}$, concatenating $\mathbf{x}, \mathbf{S}\mathbf{x}, \ldots, \mathbf{S}^K\mathbf{x}$ and feeding the result into an MLP. For the other models, we use the PyTorch Geometric implementations with standard hyperparameters.

**Node classification.** The final layer applies a convolution followed by Softmax.

**Graph classification.** The final layer applies a convolution followed by a global add pooling operation to aggregate node features into a graph-level embedding. Then, it is followed by Softmax.

## H  THE USE OF LARGE LANGUAGE MODELS

During the preparation of this work, the authors used ChatGPT to assist with grammar checking and text polishing. After using this tool, the authors carefully reviewed and edited the content as needed and take full responsibility for the content of this publication.

