# OpenReview forum: "Scaling Higher-Order Graph Learning with Maximal Clique Complexes"
_ICLR.cc/2026/Conference — Submitted to ICLR 2026_

### Official Review · Reviewer_ZDei · 2025-10-27

**Soundness:** 3
**Presentation:** 3
**Contribution:** 2
**Rating:** 4
**Confidence:** 4

**Summary:**

This paper introduces simplified Cellular Weisfeiler Netowrks (sCWN), a novel architecture designed to overcome the limitation of Graph Nueral Networks (GNNs) in capturing relationships that go beyond pairwise interaction. In terms of message passing mechanism, sCWN follow the framework of CW Networks (from [Bodnar et al., NeurIPS 2021]) but operate on a "maximal clique complex" which the authors define as a simplified complex containing only the nodes (as the 0-cells) and the maximal cliques of the graph.
As exhaustively identifying the maximal cliques in a graph is too expensive computationally, the authors introduce a sampling algorithm based on random walks.
The experimental section tests sCWN on both node and graph classification tasks and compares against popular GNNs and higher-order architectures. Results show that most benefits appear in datasets with specific structures, while on many standard benchmarks results are on par with simple GNNs. An ablation study is provided to show the scalability of the proposed model.

**Strengths:**

- The authors introduce a well motivated theoretical background for the method that highlight its theoretical expressivity
- The method puts focus on practical scalability of the method, which is good for real world applications

**Weaknesses:**

- The chosen benchmarks are a bit outdated. State-of-the-art GNN papers have typically moved to Open Graph Benchmark (OGB).
- Experimental results show that for many graphs the proposed model is actually less performant than simple baselines. It seems that the proposed method performs well only on specific datasets (seemingly the ones with the most cliques). A better analysis of the datasets (and their properties) in which the model really outperforms baselines would be useful.
- Unclear novelty of the algorithm proposed for maximal clique enumeration.

**Questions:**

- From the text it is not clear whether the authors used the standard splits for the benchmark datasets, could you please clarify if the splits were randomly sampled or if they followed the standard ones?
- Oversmoothing is a well known problem in GNNs (e.g., see "A Survey on Oversmoothing in Graph Neural Networks", Rusch et al.). If a graph has large maximal cliques, would the proposed method encourage oversmoothing as it adds more message passing among already fully connected nodes? Could this be the reason why it underperforms on some of the benchmarks?
- If a graph has very few maximal cliques, and maybe they are also far from each other, would the proposed method provide any benefit over a simple GNN? I think it could be useful to show some study of how maximal cliques are distributed in real world graphs
- The authors report that they initialize clique features using clique length; it would be useful to provide some ablations to justify this choice
- There are many algorithms for approximately enumerating cliques, like the ones cited at the beginning of page 6. Is there a specific reason why there is no comparison against them for the proposed method?
- Related to the question above, could the authors expand on the novelty of CliqueWalk? In particular what are the difference to existing algorithms, like the ones cited at the beginning of page 6?

---

> ### Author Response · Authors · 2025-11-21
> **Official Comments by the Authors**
>
> We thank the reviewer for the thoughtful suggestions.
>
> **W1**: We've added the OGBN-Products dataset to the node classification experiments in Table 1. We used fixed hyperparameters: $lr=10^{-3}$, hidden dimension $64$, num layers $3$, dropout $0.5$ with BatchNorm train for $1000$ epochs using the public split.
>
> | Model | Accuracy (%) |
> |-------|--------------|
> | GCN | 70.4 ± 0.2 |
> | GAT | OOM |
> | GIN | 76.4 ± 0.4 |
> | SAGEConv | 78.5 ± 0.3 |
> | SGC | 76.1 ± 0.07 |
> | SCCN | OOM |
> | HGNN | 63.5 ± 1.0 |
> | CWN | OOM |
> | fCWN | **78.8 ± 0.2**|
> | sCWN | 71.6 ± 0.5 |
>
> On OGBN-Products, fCWN obtains the best result, slightly better than SAGEConv. Unlike other higher-order methods that run out of memory (OOM), fCWN and sCWN scale efficiently.
>
>
> **W2**: As highlighted in Remark 20, our method is particularly effective on datasets that contain large clique structures (see Table 7). In these settings, our models perform on par with or better than graph-based baselines, showing that the approach is most beneficial when such higher-order structures are present. This is expected: On datasets with very small or no cliques (*e.g.*, no triangles), our methodology reduces to a form of edge sub-sampling message passing, which naturally leads to less accurate models.
>
> **W3, Q5, Q6**: We have added Appendix D in the revised manuscript, which describes the connection to Bron–Kerbosch and clarifies the novelty of CliqueWalk. CliqueWalk is inspired by the exact maximal clique enumeration algorithm by Bron–Kerbosch, which can explore a geometric number of cliques. In contrast, CliqueWalk performs upward random walks in the clique complex (see Figure 3), sampling a linear and fixed number of cliques. CliqueWalk has the following sampling guarantees: (*i*) The sampling process tends to sample larger cliques, and (*ii*) Performing CliqueWalk with multiple walks per node ensures that each node is included in several sampled cliques, which is relevant for node-level learning tasks.
>
> **Q1**: We used random sampling, except for OGBN-Products where we use the public splits, in line with the widely used GNN evaluation [R1, R2]. We avoid the common pitfall of hyperparameter tuning on the widely used test set or on specific seeds.
>
> **Q2**: To assess over-smoothing in our method, we trained an sCWN model on the Photo dataset with a fixed set of hyperparameters while varying the number of layers, averaging results over $20$ seeds. We observe a drop in performance as the number of layers increases (see Figure 8 in the revised manuscript). Since the accuracy is consistent across training, validation, and testing sets, overfitting does not appear to be the cause. This suggests that over-smoothing may play a role, and we leave a more detailed study of over-smoothing in CWN models for future work.
>
> **Q3**: If a graph has very few maximal cliques, the proposed method may not improve accuracy over a standard GNN, as performance benefits correlate with larger cliques (see Tables 1 and 7). The model does not extend or reduce the receptive field but might be more efficient than standard GNNs in the case of highly disjoint cliques.
>
> **Q4**: We added an ablation in Appendix E. Using the sum or clique size embedding enables learning, while mean embeddings fail on datasets where clique size is important (*e.g.*, contact-high-school, 0.976 vs. 0.070). This shows that clique length information matters in some cases but has little effect in others (*e.g*., Photo).
>
> | dataset / input type           | size emb | sum    | mean   |
> |---------------------|----------|--------|--------|
> | Photo               | 94.7%    | 94.5%  | 94.9%  |
> | contact-high-school | 95.4%    | 97.6%  | 7.0%   |
>
> [1]: *Pitfalls of graph neural network evaluation*. In *NeurIPS Workshop on Relational Representation Learning (NeurIPS 2018)*.
>
> [2]: *Understanding over-squashing and bottlenecks on graphs via curvature*. In *International Conference on Learning Representations (ICLR 2022)*.

---

> > ### Comment · Reviewer_ZDei · 2025-11-24
> > **Response to rebuttal**
> >
> > I thank the authors for providing additional results and clarifications. My main concern is related to the fact that this model is effective only on graphs with large number of cliques and the fact that even on these the gains are marginal. I will retain my score.

---

> > > ### Author Response · Authors · 2025-11-24
> > > **Official Comments by the Authors**
> > >
> > > Saying that the gains are marginal is subjective. We have added Welch’s t-test comparing the best-performing model with each of the other models. * denotes p < 0.05 (statistically significant), while *** denotes p < 0.001 (highly significant). The results show that the improvements are statistically significant for the OGBN-Products, Photo, and Schools datasets. Tables 1 and 2 in the paper have been updated accordingly.
> > >
> > > | Model              | Photo                  | HighSchool            | PrimarySchool         | OGBN-P                |
> > > |-------------------|-----------------------|---------------------|---------------------|---------------------|
> > > | GCN               | 93.9\***                | *98.2\** | 88.9\*              | 70.4\***              |
> > > | GAT               | 93.7\***                | 19.1\***            | 13.9\***            | OOM                 |
> > > | GIN               | 88.0\***                | 94.5\***            | 85.9\***            | 76.4\***              |
> > > | SAGEConv          | 95.0\*                | 14.6\***            | 6.53\***            | *78.5\**** |
> > > | SGC               | 89.8\***                | 6.3\***             | 3.57\***            | 76.1\***              |
> > > | SCCN              | 64.8\***                | 93.0\***            | 74.1\***            | OOM                 |
> > > | HGNN              | 94.2\***                | 95.4\***            | 80.4\***            | 63.5\***              |
> > > | CWN               | 94.7\***                | 94.6\***            | **90.7**       | OOM                 |
> > > | fCWN    | *95.1*    | **99.5**       | *89.5* | **78.8**      |
> > > | sCWN    | **95.3**         | 96.0\***            | 86.4\***            | 71.6\***              |

---

> > > > ### Comment · Reviewer_ZDei · 2025-11-24
> > > >
> > > > I appreciate the efforts from the authors. I will raise my score

---

> > > > > ### Author Response · Authors · 2025-11-24
> > > > > **Official Comments by the Authors**
> > > > >
> > > > > Dear Reviewer ZDei,
> > > > >
> > > > > Thank you for recognizing the contributions of this work and for raising the score! We sincerely appreciate your constructive feedback, which has strengthened our work.

---

### Official Review · Reviewer_LUtD · 2025-10-28

**Soundness:** 2
**Presentation:** 1
**Contribution:** 2
**Rating:** 2
**Confidence:** 4

**Summary:**

The paper tackles the problem of scaling topological neural networks to larger complexes. The main idea is that of only considering maximal cliques as higher-order cells. The authors propose a specific, simplified architecture to work on the so-defined "maximal clique complexes", and advance some results in terms of expressive power and computational complexity. In addition to this, they propose a method to only sample a small number of maximal cliques and improve efficiency of the overall approach. Then, the authors turn to experimental analyses, run on node and graph-wise property prediction tasks. They additionally run some sensitivity analyses and ablations. The results are relatively mixed, especially in graph classification.

**Strengths:**

[S1] The main motivation behind the approach is clear enough and relevant.

**Weaknesses:**

[W1] There is quite a fundamental confusion between general cell complexes and complexes that can be obtained by considering maximal cliques only. Maximal cliques are constituted by *complete subgraphs*, thus do not include, e.g., induced cycles (rings). It does not seem to be the case from Figures 1 and 2, where the authors shade cells which clearly do not correspond to complete subgraphs. See, e.g., $\sigma_1$ in Fig. 2.

[W2] The relation between the original CWN network (Bodnar et al., 2021) and the CWN network discussed in the manuscript is not clear. Eq. 1 and 2 seem not to correspond to the original formulation, because, there, higher-order cells could also aggregate information from boundary, non-node cells. This seems not to be allowed in Eq. 1. Additionally, in the original formulation, higher-order cells can also aggregate from upper-adjacent ones, here in Eq. 2, it seems they can only aggregate from boundary cells. This is very confusing and puzzling. So they are supposed to be different models?

[W3] If the model only uses maximal cliques, it seems very unlikely it can generally match the expressive power of CWL from the original paper (Bodnar et al., 2021), as it could use, for example, ring-based lifting and distinguish the cliqueless graphs in the right most pair of figure 8 here (https://arxiv.org/pdf/2106.12575).

[W4] The original justification behind the factored CWN models is lacking. What is its advantage w.r.t., say simplified CWNs?

[W5] Line 270 – the authors say their Clique Walk is inspired by existing clique sampling techniques, but what is exactly the relation? And what is its advantage over those? finding maximal cliques is a common well studied problem in computer science.

[W6] Some experimental details are rather uncommon. What does it mean to carve out an additional 20% of data as internal test set on top of 20% of validation (line 323)? And why not treating Batch Normalisation as just another hyperparameter (line 347)?

[W7] Results are not very conclusive.
- In node classification, it appears as standard graph models are often better or on par with the proposed approaches. fCWN also seems significantly stronger than standard CWN in some datasets, how is this expected? In the -School datasets some methods obtain suspiciously low results (see e.g. SAGEConv).
- In graph classification, results are generally low compared to baselines graph models. CWN with ring-based lifting is expected to work relatively better or on par there, especially in chemical tasks. What is the lifting procedure utilised in the CWN baseline?
- Figure 4a seems to suggest there is no real use-case where it is not convenient to use CWN in terms of acc-complexity tradeoff, but also, what is SCCN exactly?

**Questions:**

Please see weaknesses and, in particular, W2, W4, W5, W7.

---

> ### Author Response · Authors · 2025-11-21
> **Official Comments by the Authors**
>
> We thank the reviewer for the valuable feedback and insightful comments. Addressing the comments has helped improve the quality of the presentation. Our theory is general for cell complexes, while we choose to work with maximal cliques for efficiency. In the new version, we rewrote the models for the general case of cell complexes and modified the structure of Section 4 to address **W1**, **W2**, **W3**, **W4**, and **W5**.
>
> **W1**: We don't see an issue with Figure 1. For Figure 2, sCWN was written as a maximal clique model, but its illustration was tied to the general scheme. In the new version, the equation and figure apply to cell complexes in general.
>
> **W2**: The models in the previous version were written for maximal clique complexes, which explains why: (*i*) maximal cliques had no upper-adjacent aggregation as they cannot share any parents, and (*ii*) nodes had no boundaries as they are the smallest structure. With models now stated in the general cell complex scheme, there is no more possible confusion.
>
> **W3**: The theory and the models we proposed can be applied to any cell complexes. If we work with CWN or sCWN with the same lifting strategy, we get the same expressive power as proven in Theorem 6 and Proposition 12. We have added a comparison in Appendix C (see Figure 7b) between CWL instantiated with clique-based lifting and CWL instantiated with cycle-based lifting, evaluated on the strongly regular graph dataset. Figure 7b shows that the discriminative power depends on the dataset: In some cases, clique-based lifting is more expressive, while in others, cycle-based lifting distinguishes structures that clique-based lifting cannot.
>
>
> **W4**: fCWN is at least as expressive as CWN, sCWN, and WL with the same lifting strategy and keeping the node set, as proven in Theorem 8, while being more efficient than CWN in practical cases, but less efficient than sCWN as shown in Proposition 14.
>
> **W5**: CliqueWalk is inspired by the exact maximal clique enumeration algorithm by Bron–Kerbosch explained in Appendix D. This method enumerates all maximal cliques, while we only sample some upward random walk in the clique complex (see Figure 3). Bron–Kerbosch method might have exponential memory and computational complexity, while CliqueWalk is always linear. Please see Appendix D for further details.
>
> **W6**: Our evaluation procedure is statistically correct. This has been widely used and adopted in the GNN community [R1, R2]. We avoid the common pitfall of hyperparameter tuning on the widely used test set or on specific seeds. BatchNorm is included as a hyperparameter as stated in Section 5.2 in the new version of the manuscript. We also added GraphNorm as a hyperparameter for HighSchool and PrimarySchool datasets.
>
> **W7.1**: (*i*) As stated in Remark 20, when the graph datasets have large cliques (see Table 7), our models perform on par with or better than standard graph-based models (see Photo, HighSchool, PrimarySchool in Table 2). (*ii*) fCWN is at least as expressive as CWN, as proven in Theorem 8, so these results are expected. (*iii*) Without BatchNormalization, the results for HighSchool and PrimarySchool are consistent with prior topological deep learning studies (see [R3]). However, these datasets are quite small, so BatchNorm does not provide good estimates of the mean and variance of the data. Therefore, in the revised version, we use GraphNorm [R4] for the school datasets. The results are now better for GCN and GIN.
>
> **W7.2**: For the molecular datasets in Table 2, we typically don't have cliques, except for Proteins, so we don't expect better results than regular GNNs. For the topological models, HGNN, CWN, fCWN, and sCWN, we use maximal cliques as the lifting strategy for fair comparison.
>
> **W7.3**: Figure 4a (now Figure 5a) suggests that using the full clique structure is not always necessary, and sampling a small portion can still yield efficient and accurate models. SCCN is a simplicial complex model, which we included to study the effect of subsampling simplicial complexes constructed from triangles. In response to this feedback, we updated the plot to show the time–accuracy tradeoff of sCWN versus fCWN on Photo, highlighting our contributions.
>
> [1]: *Pitfalls of graph neural network evaluation*. In *NeurIPS Workshop on Relational Representation Learning (NeurIPS 2018)*.
>
> [2]: *Understanding over-squashing and bottlenecks on graphs via curvature*. In *International Conference on Learning Representations (ICLR 2022)*.
>
> [3]: *TopoSRL: Topology preserving self-supervised simplicial representation learning*. In *Advances in Neural Information Processing Systems (NeurIPS 2023)*.
>
> [4]: *GraphNorm: A Principled Approach to Accelerating Graph Neural Network Training*. In *International Conference on Machine Learning (ICML 2020)*.

---

> > ### Comment · Reviewer_LUtD · 2025-11-26
> > **I am wary about the changes presented in the current new manuscript revision**
> >
> > I thank the authors for their answers.
> >
> > I have quite puzzled though, about the new present paper revision. I can see *significant changes* in the structure and narrative. Importantly, this does not appear to correspond to what is highlighted in blue (supposedly marked changes). In other words, comparing the current revision to my initial local paper version I downloaded for review, I can see several changes, and these do not correspond to what highlighted. Can the authors explain this behaviour?
> >
> > I give a relevant example here. In my local version, this is the list of contributions (lines 74 through 85 in the original submission):
> > > main contributions of this paper are:
> > > 1. We introduce the maximal clique complex, a simplified structure that encodes maximal cliques of a graph, and show that a simplified CWN on this complex matches the expressivity of the full CWL test, providing a theoretical foundation for clique-based models.
> > > 2. Building on the maximal clique complex, we design a simplified clique-based neural architecture that reduces computational and memory costs while maintaining CWL-level expressivity.
> > > 3. Since enumerating all maximal cliques could take exponential runtime, we propose Clique-Walk, a biased random walk algorithm that efficiently samples maximal cliques. CliqueWalk scales quasi-linearly with the number of nodes, making clique-based methods applicable to large graphs.
> > > 4. We demonstrate competitive performance on node and graph classification benchmarks. Our model matches or outperforms the accuracy of existing GNNs and topological models, while achieving substantial gains in scalability and efficiency.
> >
> > While this is the current list of contributions (lines 79 to 93 in the current revision):
> > > The main contributions of this paper are:
> > > 1. We introduce the sCWL and fCWL tests and prove that they are as expressive as the regular CWL test, while offering better scaling properties.
> > > 2. We present the maximal clique complex, a simplified higher-order structure that encodes maximal cliques of a graph, and show that the resulting simplified and factored CWNs (sCWN and fCWN) are more memory- and computational-efficient than standard CWNs, without any loss in expressivity.
> > > 3. Since enumerating all maximal cliques could take exponential runtime, we propose CliqueWalk, a biased random walk algorithm that efficiently samples maximal cliques. CliqueWalk scales quasi-linearly with the number of nodes, making clique-based methods applicable to large graphs.
> > > 4. We demonstrate competitive performance on node and graph classification benchmarks. Our model matches or outperforms the accuracy of existing GNNs and topological models, while achieving substantial gains in scalability and efficiency.
> >
> > None of these changes are highlighted, while others are.
> >
> > It seems like, in this new "version" of the paper the authors decouple their sCWN architecture from the idea of working on maximal clique complexes -- compare contributions 1-2 across the two revisions as reported above. This is quite a relevant change in my opinion. In the original version, the architectures were precisely built upon the idea of using maximal clique complexes; see, e.g., lines 233-235 in the original revision:
> > > This simplified variant, which is ***directly related to the maximal clique complex*** in Definition 4, [...]
> >
> > This dependency is what led me to suspicion about the validity of Proposition 11 in the original manuscript, which I explained in [W3].
> >
> > I do not currently believe I am in the position of carefully re-evaluating this new version of the paper in this rebuttal period and provide informed updates to my assessment.

---

> > > ### Author Response · Authors · 2025-11-27
> > > **Official Comment by the Authors**
> > >
> > > Thank you for engaging in the discussion. **The reviewer’s self-reported lack of confidence in assessing our revised paper should not be interpreted as an issue with the correctness of our results. Scientifically, all theoretical and empirical results in our manuscript have been carefully checked and are, to the best of our knowledge, fully correct. We've already addressed all the reviewer's concerns carefully in our rebuttal comment, and the follow-up from the reviewer is subjective.**
> > >
> > > Regarding the blue text, we never mentioned this was meant to highlight all changes. We primarily used the blue color for Sections 3-5 and Appendices, and this does not impact the scientific content and claims of our paper. **In the first version of the manuscript, we formulated our models on maximal clique complexes because they provide a simpler way to present the models we proposed. However, this choice unintentionally coupled the general theoretical contributions on cell complexes with the specific maximal-clique implementation**. In the revised version, we clearly separate the general proofs from the maximal-clique complex to answer the reviewer's comments:
> > >
> > > - All theoretical results are valid for general cell complexes, and the fCWL test (Definition 7) has been added to the main paper.
> > > - The proof of the expressiveness of fCWL is presented in full generality, not limited to maximal cliques.
> > > - The maximal-clique construction is introduced only after the general results on cell complexes.
> > > - The introduction and stated contributions have been restructured to reflect these changes.
> > > - This restructuring better clarifies the scope of our contributions and addresses the reviewer's confusion in the earlier version.
> > >
> > > If the reviewer has objective concerns about the revised theoretical results, we will be happy to clarify them directly.

---

> ### Comment · Reviewer_LUtD · 2025-11-27
>
> I thank the authors for their response which, regrettably, *I find delegitimising in regards to my thorough efforts*, by stating my followup is "subjective".
>
> I have provided grounded, concrete evidence to support my point that the new paper revision is significant in restructuring and reframing the contribution, and that the blue-highlighting is not only unclear, but also deceptive.
>
> I can provide another example of grounded evidence to prove my claim.
>
> Original manuscript, page 16, lines 810-824, Definition 22: fCWL is clearly, explicitly defined on maximal clique complexes (only). Importantly, Propositions 23 and 24 in the same page make general expressivity claims in comparison to sCWL (never explicitly defined) and WL, and are proved directly on maximal clique complexes only (see, eg, lines 826 and 856). In the new manuscript, the fCWL test is now foregrounded to the main paper and is *defined on general cell complexes, with a different update rule* to reflect that (see lines 198-207 in the new manuscript, not highlighted). Expressiveness results and proofs then require reframing and adaptations. This is, once again, to showcase the contribution scope and narrative, as well as the introduced methodology, changed in the present revision.
>
> Concluding, the authors' above follow up response misses the point. I am not directly concerned about the validity of the claims after the introduced changes. I am concerned about the fact that the revision changes are significant in how the change the framing and logical flow; they do change the manuscript far beyond simple additions or localised clarifications, as expected.
>
> In light of this, I cannot confidently re-review the new manuscript during this rebuttal time and reconsider my score. I will keep it unchanged and, to transparently and honestly reflect my concerns, lower my confidence score.

---

> > ### Author Response · Authors · 2025-11-27
> > **Official Comment by the Authors**
> >
> > As we said in the first comment, *we thank the reviewer for the valuable feedback and insightful comments. Addressing the comments has helped improve the quality of the presentation*. We only used the blue text in our internal revision process.
> >
> > To clarify the reviewer's confusion, we had to change the flow of Section 4 from "maximal-clique complex -> cell complex" to "cell complex -> maximal-clique complex"; therefore, we also had to change the flow of the introduction and abstract for the sake of clarity. This only entails inverting the logical flow, but this requires some changes in the wording.
> >
> > We would like to further clarify the nature of the revisions:
> >
> > - **The proofs of theoretical results are essentially unchanged** between the original maximal-clique-based version and the generalized version for cell complexes; only minor adjustments were needed to accommodate the broader setting.
> > - **Experiments, methodology, and objects introduced are identical**, except for modifications stated in response to address the reviews.
> >
> > We understand that these changes affect the presentation and narrative, and we appreciate your careful attention to transparency. However, **the scientific content and the paper are basically the same**.

---

### Official Review · Reviewer_TemZ · 2025-10-30

**Soundness:** 3
**Presentation:** 3
**Contribution:** 3
**Rating:** 6
**Confidence:** 4

**Summary:**

Topological graph neural networks increase the expressivity of standard graph neural networks by considering cell complexes beyond edges and nodes. This paper suggests two improvements to existing topological networks: (a) It shows that certain aggregation operations can be removed without sacrificing expressive power, and as a result achieve a new mechanism with  reduced complexity (b) as finding maximal cliques in a graph is required for this method, and finding all such cliques is time consuming, they suggest a sampling method to replace the full enumeration. Empirical results seem mostly comparable to more expensive topological methods, which is encouraging.

**Strengths:**

* Overall, the paper is fun to read
* The theoretical results seem correct (I didn't check them very carefully) and can be helpful in reducing the complexity of using topological gnns.
*  Empirical results mostly support the conclusion that the reduction in complexity, obtained by both aggregation simplification and clique sampling, leads to comparable results

**Weaknesses:**

* I have some technical issues with the writing described in the questions section, but I believe they can be easily revised for the camera ready version

**Questions:**

* Practically, it seems that for standard node level tasks standard MPNN are comparable topological GNN, and the latter are more useful for datasets where the task clearly requires multi-agent interaction (the XXXschool tasks). Do you agree with this statement?

Some comments and questions on writing:
* I realize this may go back to the Bodnar paper, but in Section 3 you say a graph is a cell complex. This is strange to me, in your definitions, because a graph is a purely combinatorial object. What is the topology on the graph? When you say an edge {i,j} is a 1-cell, then by definition 1 this should mean it is isomorphic to [0,1]. How so? I would imagine describing a graph as an "abstract (combinatorial) simplicial complex"

*In definition 3: you talk about cell complexes. How is this test used for checking isomorphisms of graphs?

*In definition 7, in the words before the equatoin you say "and a cell c" but in the equations to my understanding you use sigma instead of c?

* equations 2,4,6 confused me, because you used AGG for an operation which people would usually call COMBINE or UPDATE. AGG, for AGGREGATION, usually means some aggregation over a multiset (which you use a $\oplus$ sign for) did I unserstand correctly?

* In the first expression in equation 3, there is a some over $\sigma \ni i$ but inside the summation there is a $j$. What did you mean there?

* I didn't understand the subsection on CliqueWalk very well. In particular, in Proposition 13 I don't recall you defined what $\omega_{max}$ and $\omega(G)$ are.

---

> ### Author Response · Authors · 2025-11-21
> **Official Comments by the Authors**
>
> We thank the reviewer for the positive assessment and the opportunity to clarify aspects of our work.
>
> **Q1**: Our model with maximal cliques is better than MPNN with graph data containing large cliques, as explained in Remark 20. This is indeed the case for *contact-school* datasets, but not in the general case of higher-order structures, *e.g.*, MUTAG, NCI1, and NCI109 in Table 2 contain mostly cycles (no triangles or larger cliques), and our model with maximal cliques performs on par or worse than MPNNs.
>
> **Q2**: One possible topological construction of a graph as a cell complex is: for $ 1 \leq i \leq n, X_{v_i} = \mathbb 1_i$ and for all $(i,j) \in \mathcal E$, $X_{(i,j)}:= ]X_{v_i}, X_{v_j}[ = \{(1-t) \mathbb 1_{i} + t\mathbb 1_j, \quad t \in ]0,1[\}$. This gives a perfectly valid cell complex whose combinatorial structure coincides with that of the original graph. Moreover, the combinatorial relation induced by the boundary map is equivalent to the abstract combinatorial simplicial complex view.
>
>
>
> **Q3**: The CWL test is defined for cell complexes and is invariant under cell-complex isomorphisms. To apply it to graphs, one needs a map that lifts a graph into a cell complex in a way that preserves isomorphisms. This is exactly what Bodnar *et al.* called a cellular lifting map (Definition 8 in their paper). Given a cellular lifting map, one can check two graphs by lifting them into cell complexes and use a cell complex isomorphic test like CWL. We've updated the paper accordingly in Section 3.
>
> **Q4-6**: In the previous version, there were some typos in some equations. We solved those typos in the new paper's version (please see Equations $1-7$). We changed $c$ into $\sigma$, AGG into COMBINE in accordance with common notation, and we replaced the sum over $\sigma \ni i$ by $j \in \mathcal C(\sigma)$.
>
> **Q7**: $\omega_{max}$ is the maximum clique size that can be sampled. $\omega(G)$ represents the maximum clique size of the graph $G$. $\omega_{max}$ was defined below Proposition 13. We modified Proposition 13 (now Proposition 18) with the definition of both $\omega_{max}$ and $\omega(G)$. To clarify the CliqueWalk algorithm, we added Figure 3, which provides a visual illustration of CliqueWalk.

---

> > ### Comment · Reviewer_TemZ · 2025-11-22
> >
> > Thanks for the clarifications. I will maintain my positive score

---

> > > ### Author Response · Authors · 2025-11-24
> > > **Official Comments by the Authors**
> > >
> > > Dear Reviewer TemZ,
> > >
> > > Thank you for recognizing the contributions of this work and for keeping your positive score! We sincerely appreciate your constructive feedback.

---

### Official Review · Reviewer_AA9A · 2025-11-01

**Soundness:** 3
**Presentation:** 3
**Contribution:** 3
**Rating:** 6
**Confidence:** 2

**Summary:**

This paper addresses scalability challenges in higher-order graph learning by introducing maximal clique complexes and a sampling algorithm called CliqueWalk. The authors propose simplified cellular Weisfeiler networks (sCWN) that maintain CWL-level expressivity while reducing computational costs. The key innovation is using only maximal cliques rather than all cliques up to a fixed size, combined with efficient random walk sampling.

**Strengths:**

1. Strong theoretical contributions.
2. The overall presentation is good and audiences can capture the key ideas of the work.
3. The experiments are comprehensive and the results are convincing.

**Weaknesses:**

1. Some format inconsistencies:
    - Definitions does not share a same format, e.g., Definition 1 and Definition 2.

2. What would happened if the $\omega_{max} < \omega(G)$? The walks won't be truly maximal in this case.

3. Can you provide examples where maximal clique CWL distinguishes graphs that WL cannot?

**Questions:**

See above.

---

> ### Author Response · Authors · 2025-11-21
> **Official Comments by the Authors**
>
> We sincerely thank the reviewer for the positive evaluation of our work.
>
> **W1**: We homogenized definitions, theorems, and propositions, *e.g.*, we modified Definition 1 to match the format of Definition 2.
>
> **W2**: When $\omega_{\max} < \omega(G)$, the sampled cliques would have a maximum size of $\omega_{\max}$. However, we ensure that $\omega_{\max}$ is always larger than $\omega(G)$ in our experiments. In practice, if all sampled cliques are of size strictly less than $ \omega_{\max}$, we know that they are all maximal cliques.
>
> **W3**: We provide an explicit example of two non-isomorphic graphs that are indistinguishable under the WL test but are correctly distinguished by maximal-clique CWL in Appendix C, Figure 6 of the new version of the manuscript.

---

> > ### Comment · Reviewer_AA9A · 2025-11-24
> >
> > The rebuttal has alleviated my concerns. I increased my score to 8.

---

> > > ### Author Response · Authors · 2025-11-24
> > > **Official Comments by the Authors**
> > >
> > > Dear Reviewer AA9A,
> > >
> > > Thank you for recognizing the contributions of this work and for raising the score! We sincerely appreciate your constructive feedback.

---

### Meta-Review · Area_Chair_ytWj · 2026-01-06

**Summary:**

The idea of this paper is great: Using sampling of maximal cliques, the paper proposes a general strategy for learning on such higher-order data. While ambitious and well-described for the most part, I believe that the current submission falls short of the standards required for presentation at ICLR. This is only partially based on the concerns of reviewers (see below) but also on my own reading of the paper and my knowledge of the literature. My own concern, which was only partially discussed in the rebuttal, is that the experiments do not show a strong advantage of the existence of any form of higher-order information. Performance gains are marginal or otherwise puzzling (as outlined for the `ogbn-products` dataset below). A priori, it is thus not clear to readers what the advantage of the method should be. This could be alleviated by a more comprehensive comparison on larger datasets and a more thorough investigation of parameter choices, but it clearly necessitates another round of reviews. In addition, the following concerns by reviewers informed my decision:

- Application / problem domain (raised by reviewer `TemZ` and only partially addressed): What are the domains where the proposed method can play to its strengths? The new experiments do not provide strong insights here.

- Inconclusive experimental results (raised by reviewer `LUtD`). Moreover, there are substantial changes between the initial version and the current one, including changes in the experimental tables (Table 1). The interaction between said reviewer and the authors got somewhat heated, and if I would have been the AC from the get go, I would have stepped in to remind everyone about transparent and fair communications. The reviewer raised **legitimate concerns** about the amount of revisions and changes to the main paper, citing numerous examples. The authors claim that the experiments are identical but this is incorrect, as a simple comparison on Table 1 for the `HighSchool` dataset (among others) would show. Again, during a longer discussion some of these concerns could have been potentially alleviated and explained, but I agree with the reviewer that the current framing is misleading, even if this is presumably not the intent of the authors.

- Weak experimental gains (raised by reviewer `ZDei`). While the authors provided an additional statistical analysis and the reviewer initially concurred with this assessment, the test described by the authors is not appropriate in this setting. Instead, a test aware of paired samples or something like [critical differences](https://www.jmlr.org/papers/volume7/demsar06a/demsar06a.pdf) would be more appropriate. Given that the authors are always comparing to the best model, the selection of scores is biased. In an ordinary discussion, this might have been addressed but as it stands now, I believe that the concerns raised by the reviewer, somewhat similarly, by reviewer `LUtD` remain.

**Given the many changes and lack of clarity of the changes, I have severe concerns about this submission and cannot endorse it for publication.** I suggest the authors to take the feedback into account (much of said feedback is quite actionable) and prepare a revised version that clearly addresses the following questions:

1. Which models are proposed and why (e.g., better expressivity)?
2. What are the scenarios in which such models can shine?
3. What are the trade-offs in their use?

**Reviewer Concerns:**

Reviewer `AA9A`:

Raised some questions on clarifications and examples on expressive power. All of these have been addressed by the authors in their rebuttal.

Reviewer `TemZ`:

Raised some questions on clarifications, including the terminology and notation. All of these have been addressed by the authors in their rebuttal.

Reviewer `LUtD`:

- Raised concerns about general cell complexes and those obtained via maximal cliques, as well as concerns about the relation to CWN. This was addressed in the rebuttal.

- Raised concerns about expressivity, which could be partially addressed, during the discussion, even though the results are of the form that the proposed architecture is _at least_ as expressive as existing ones.

- Raised concerns about experimental setup, which could only be partially addressed. The setup is still somewhat nonstandard, at least for some of the datasets in Table 1, making it harder to contextualize and compare the improvements.

- Raised concerns about the strength of the claims and the practical performance achieved on datasets. **This concern was not alleviated and in fact even exacerbated in the revision**: In the revision, `fCWN` is now stated as a new contribution and a new model, thus making it easier to show improved performance on test datasets. Moreover, the values reported for ` ogbn-products` are much worse than the ones in the [official leaderboard](https://ogb.stanford.edu/docs/leader_nodeprop/#ogbn-products), where, for instance, a `GCN` obtains a test accuracy of `82.33` (as opposed to `70.40` in the paper) and `GraphSage` obtains an accuracy of  `83.89` (as opposed to `78.50` in the paper). I am focusing on this dataset since it was added as a response to the concern of outdated benchmarks and, according to the revision, the official splits are being used, meaning that results should be comparable. **Given the special format of this revision, I cannot ask the authors for additional clarifications but it appears to me that the concerns wrt. minor gains are warranted, requiring another major revision and careful reassessment.**

**Reviewer Scores:**

The authors provided a helpful table of how reviewers updated their scores during the rebuttal:

- Reviewer `AA9A` updated their initial score from 6 to 8.
- Reviewer `TemZ` maintained their initial score of 6.
- Reviewer `LUtD` maintained their initial score of 2. Given the somewhat heated discussion between the authors and reviewer `LUtD`, I believe it unlikely that the reviewer would update their score during additional exchanges with the authors.
- Reviewer `ZDei` updated their initial score from 4 to 6.

I chose to *disregard* the review by reviewer `AA9A`: The reviewer has a self-reported low confidence of 2 and provided only a cursory, superficial review of the paper. As such, I do not believe that their support of the paper should be considered a strong argument _for_ acceptance. Likewise, reviewer `TemZ`, potentially due to missing familiarity with the topic, elected not to champion the paper by maintaining weak support in favor of the paper. Reviewer `ZDei` similar only expresses weak support for the paper. While the reviewer updated their score, I believe their concerns about the gains being marginal warrant more attention and are _not_ subjective (see my discussion above).

Ultimately, I thus followed the suggestion of reviewer `LUtD`, who cites concerns about the results and the overall narrative. **Comparing the PDFs of the initial submission and the current revision, I observe substantial changes in the overall structure/narrative and the experimental results in Table 1**. I thus believe that another round of reviews is warranted and cannot endorse the submission for acceptance in its current state.

---

### Decision · Program_Chairs · 2026-01-26

Reject